

# The Fate of a Southwest Pacific Bloom: Gauging the impact of submesoscale vs. mesoscale circulation on biological gradients in the subtropics

Alain de Verneil[1], Louise Rousselet[1], Andrea M. Doglioli[1], Anne A. Petrenko[1], and Thierry Moutin[1]

[1]Aix Marseille Univ, Université de Toulon, CNRS, IRD, OSU PYTHEAS, Mediterranean Institute of Oceanography MIO, UM 110, 13288, Marseille, Cedex 09, France

*Correspondence to:* Alain de Verneil (alain.de-verneil@mio.osupytheas.fr)

**Abstract.** The temporal evolution of a surface chlorophyll-*a* bloom sampled in the Western Tropical South Pacific during the 2015 Oligotrophy to UlTra-oligotrophy PACific Experiment cruise is examined. This region is usually characterized by largely oligotrophic conditions, ie low concentrations of inorganic nutrients at the surface and deep chlorophyll-*a* maxima. Therefore, the presence of a surface bloom represents a significant perturbation from the mean ecological state. Combining in situ and
remote sensing datasets, we characterize both the bloom's biogeochemical properties as well as the physical circulation responsible for structuring it. Biogeochemical observations of the bloom document the bloom itself, a subsequent decrease of surface chlorophyll-*a*, significantly reduced surface phosphate concentrations relative to subtropical gyre water farther east, and a physical decoupling of chlorophyll-*a* from a deep nitracline. All these characteristics are consistent with nitrogen fixation occurring within the bloom. The physical data suggest surface mesoscale circulation is the primary mechanism driving
the bloom's advection, whereas balanced motions expected at submesoscales provide little contribution to observed flow. Together, the data provide a narrative where subtropical gyre water can produce significant chlorophyll-*a* concentrations at the surface that is stirred, deformed, and transported great distances by the mesoscale circulation. In this case, for the time period considered the transport is in an easterly direction, contrary to both the large-scale and mean mesoscale flow. As a result, future studies concerning surface production in the region need to take into account the role complex mesoscale structures play in
redistributing subtropical gyre water.

## 1 Introduction

Subtropical gyres and their surroundings represent the largest surface biological provinces in areal extent. These regions are characterized by low standing stocks of phytoplankton biomass with deep chlorophyll-*a* (chl-*a*) maxima (DCM's) and low surface nutrient concentrations, and so have been dubbed the oligotrophic deserts of the sea. The South Pacific gyre (SPG) is
the world's largest gyre and the most remote from large landmasses. Due to its remoteness, in situ data are generally lacking from the region; however, previous studies corroborate the oligotrophic status of this region (Claustre et al., 2008). Among oligotrophic areas, the SPG appears to be a low chlorophyll area with low $N_2$ fixation rates and high residual phosphate suggesting biological carbon pump inefficiency (Moutin et al., 2007). Flowing from the east, the waters from the SPG reach



the Western Tropical South Pacific (WTSP), which has been recently shown to be the world hotspot for $N_2$ fixation (Bonnet et al., in press), a process considered to be the largest external nitrogen source to the ocean (Sohm et al., 2011). Therefore, the physical processes that structure the gyre and nearby areas must be investigated.

Marine biological communities at any moment reflect a time-integration of the many complex interactions that occur both within the community and with the physical environment (Longhurst, 2010). Despite the constant shifting and stirring that exist in a fluid medium, investigators often espouse the assumption that near "quiescent" gyres the mean circulation's long timescale means that shipboard observations provide static, representative snapshots of a community that remains physically coherent long before and after in situ sampling. This assumption, however, is not always valid.

In recent decades, accumulating satellite data and high resolution modeling studies highlight how complex the surface ocean circulation really is beyond the mean flow, with several consequences. First, most kinetic energy in the surface ocean is found at the mesoscale, in eddies and frontal structures evolving over weeks and months (Stammer, 1997; Ferrari and Wunsch, 2009), so in a given field campaign these features can be the most important. Second, nonlinear eddy structures (ie 'rings') can be long-lived and transport water long distances (Chelton et al., 2007; Rousselet et al., 2016). These rings host their own biological dynamics, as well, which impact the biological pump (McGillicuddy et al., 2007; Nencioli et al., 2008; Moutin et al., 2012). Third, mesoscale motions can provide energy to the submesoscale through frontogenesis and filamentation, producing motions with much shorter timescales and relatively vigorous vertical circulation (Mahadevan and Tandon, 2006; Thomas et al., 2008; Mahadevan, 2016). Taken all together, these characteristics of surface ocean flow elucidate the importance of the mesoscale and submesoscale (hereafter (sub)-mesoscale when referred together) circulation in determining the spatial pattern of biological communities, with recent modeling work taking into account the role they play in moving around biological production at a regional scale (Nagai et al., 2015).

The WTSP presents an ideal laboratory in which to assess the impact of (sub)-mesoscale circulation upon planktonic communities in the gyre and nearby. In the WTSP, the SPG's northern limb, the South Equatorial Current (SEC), flows west into the Melanesian Archipelago group (MA) (Chaigneau and Pizarro, 2005). Though the SEC is relatively oligotrophic throughout, the depth of the DCM shoals as it moves west past these island groups, indicative of reduced oligotrophy. Sampling of this zonal gradient was the focus of the Oligotrophy to UlTra-oligotrophy PACific Experiment (OUTPACE) cruise during austral summer 2015 aboard the RV *L'Atalante* (Moutin and Bonnet, 2015). The main biogeochemical goals of OUTPACE concerned the study of biological production and its fate in a gradient of trophic conditions, from the ultra oligotrophic conditions of the SPG to the oligotrophic conditions of the MA (Moutin et al., this issue). During the cruise, however, a strong surface chl-*a* bloom was detected from satellite data and intensively sampled for five days during long duration station LDB. The presence of this surface bloom, itself a marked departure from the usual oligotrophic DCM pattern, in a region known to host production fueled by $N_2$ fixation at the surface, brings into sharp focus the need to quantify and contextualize the relevant (sub)-mesocale circulation, also intensified near the surface, that might influence and structure this bloom.

In this study, we use a combination of in situ and remote sensing data to describe the bloom's distribution, evaluate its temporal evolution, and what role the (sub)-mesoscale current field may play in structuring it. We outline the sources of data, their processing, and the analyses required to investigate the bloom in Sect. 2. Subsequently, Sect. 3 summarizes the results





from the calculations performed. After a discussion of the results in Sect. 4, we provide conclusions and possible applications of our approach in Sect. 5.

## 2 Materials and methods

### 2.1 Cruise sampling plan and context

In situ data for this study come from the OUTPACE cruise, conducted from 18 February to 3 April, 2015 in the WTSP on the French RV *L'Atalante*. Over a mostly zonal transect beginning west of New Caledonia and ending near Tahiti (Fig. 1), the ship sampled in two main ways: first, the short duration (SD) stations, and second the long duration (LD) stations. For the 15 SD stations, referred to by their number in chronological order, conductivity-temperature-depth (CTD) casts and biogeochemical measurements were taken over a 24 h period, as described in (Moutin et al., this issue). During the three LD stations, referred to
as stations LDA, LDB, and LDC, a quasi-Lagrangian drift array was deployed with sediment traps. The ship largely followed the trajectory of this array. Multiple CTD casts were performed, spaced every 3 h. The position of each station was chosen using the SPASSO software suite (Doglioli, 2013), which analyzes near real-time remote sensing data so that an onshore collaborator can provide daily updates regarding the location of possible coherent structures such as eddies (Doglioli et al., 2013). Station LDB was selected because of the large surface chl-*a* signal as seen from satellites (Fig. 1b). The data used specifically in this
study derived from SD12, LDB, and SD13. For these purposes, we defined the general region of interest as $184°$ E to $192°$ E and $22°$ S to $16°$ S (Fig. 1b).

  In situ sampling for SD12, LDB, and SD13 took place during 11-12, 14-20, and 21 March, 2015, respectively. Before the deployment of the drift array during LDB, a survey of the area was conducted by a Moving Vessel Profiler (MVP; Brooke Oceanographic) platform, allowing for high-resolution CTD profiles over three separate transects (T1,T2, and T3 in Fig. 2a).
Subsequent to the multi-day drift array experiments, a final MVP transect was conducted, as well (T4).

### 2.2 In situ data

#### 2.2.1 CTD, bottle data, MLD and nitracline depth

The shipboard CTD rosette was deployed four times at SD12, 47 times during LDB, and once for SD13. The rosette layout was the Seabird SBE 9+ CTD-rosette (CR), with two CTDs installed on the rosette and a chl-*a* fluorometer (Chelsea Aqua 3).
CTD data were calibrated and processed post-cruise using Sea-Bird Electronics software into 1 m bins. Chl-*a* fluorescence was calibrated to chl-*a* extractions taken from bottle samples throughout the cruise.

  Subsequent to CTD processing and calibration, each profile's density was reordered to be stable so that small, residual overturns were removed. Casts were spaced approximately 3 h apart, so density and chl-*a* were interpolated to a regular 3 h interval using cubic polynomials. A horizontal smoothing of density was performed using a 'lowess' filter with a window of
four datapoints, equivalent to 12 h. This time span, designed to remove some internal waves, is $\sim \frac{1}{3}$ of the inertial period ($\sim 36$




h), and does not remove the movements present due to near-inertial oscillations, a dominant signature in the shipboard acoustic doppler current profiler (SADCP) data (Bouruet-Aubertot et al., this issue).

Mixed layer depth (MLD) was calculated using a threshold density deviation of $0.03 \ \mathrm{kg \ m^{-3}}$ from the value at a reference depth. The CTD profiles post-calibration did not always contain surface values, so a 10 m reference depth was used, similar to
de Boyer Montégut et al. (2004).

Measurements of dissolved nitrate and phosphate concentrations were conducted for two casts at SD12, seven casts at LDB, and the single cast of SD13. The nitrate and phosphate concentrations were measured using continuous flow analysis (SEAL AutoAnalyzer 3) following the procedures in Aminot and Kérouel (2007). Quantification limits for all nutrients are 0.05 μmol $\mathrm{kg^{-1}}$. LDB nitracline depth was calculated for the last CTD associated with the drift-array recovery. The nitracline depth
($D_{NO_3}$) and its slope ($S_{NO_3}$) were calculated performing a linear fit with the first three measurements above the threshold of 0.05 μmol $\mathrm{kg^{-1}}$, resulting in $D_{NO_3} = 121$m and $S_{NO_3} = 48$ μmol $\mathrm{m^{-4}}$. The density anomaly at $D_{NO_3}$ for this cast, 24.34 kg $\mathrm{m^{-3}}$, was subsequently used for the rest of the LDB time series.

### 2.2.2 MVP

The MVP was used during four transects, three (T1-T3) before LDB and one after (T4) (Fig. 2a). Prior to LDB, we have:
T1, going Northwest to Southeast; T2, going South to North; and T3, North to South in the same path as T2. For these three transects, a total of 388 casts spanning ∼700 km distance were obtained, producing an average horizontal resolution of ∼2 km. Subsequent to LDB, T4 traveled west to east from LDB's position, resulting in 95 casts over ∼170 km horizontal distance.

The MVP vehicle sampled vertically by freewheeling the synthetic cable attached to the MVP fish, allowing for a near-vertical descent at ∼4 $\mathrm{m \ s^{-1}}$. At a prescribed depth (here, 350 dBar), the brake was applied to the computer-controlled winch
and the fish automatically brought back to the surface. Only down-casts were used in this dataset due to the deployment method.

Onboard the fish was a rapid response AML Micro conductivity sensor, thermistor, and Wetlabs Wetstar chl-*a* fluorometer. Due to technical difficulties onboard the ship, the conductivity sensor was swapped for a sound velocity sensor. In order to calculate salinity, the roots of the sound speed equation from Chen and Millero (1977) were matched with Mackenzie's linear approximation (1981). Sound speed and temperature data were lag-corrected to reduce salinity spiking. A previous study
utilizing MVP data found an operational threshold binning of ∼1m in the vertical (Li et al., 2012), which we followed here. Temperature and calculated salinity were calibrated to the station SD13 CTD cast made adjacent to and soon after the last MVP profile of T4. Chl-*a* values were calculated by calibrating to the already calibrated CTD fluorometer values for SD13, as well.

MVP density profiles, as with CTD data, were first reordered to be statically stable. In contrast to the CTD data, horizontal distance, and not time, is the relevant variable. As discussed below (Sect. 2.4), the stratification was such that, even at 2 km
resolution, density structures associated with balanced currents near the surface might be missed, so no additional filtering was applied to density in the MVP dataset despite possible aliasing of internal waves.

In order to determine when the ship was in the surface bloom, a threshold value was chosen. Inspection of chl-*a* in the upper 20 m revealed a bi-modal distribution. As a result, the value of 0.13 μg $\mathrm{L^{-1}}$, well above the lower mode and representing the 66% value of the cumulative distribution function, was selected. Therefore, by our definition the MVP entered the surface



bloom when the average value of chl-*a* in the top 20 m surpassed this value. Additionally, the MLD for the MVP transects were calculated from the same method as the CTD casts, by finding the depth where density surpasses 0.03 kg m$^{-3}$ above the value at 10 m.

### 2.2.3 SADCP

The RV *L'Atalante* has two SADCPs, RDI Ocean Surveyors with frequencies of 75 kHz and 150 kHz. In order to maximize vertical resolution, we use the 150 kHz data in this study. Single ping data were collected into 2 min intervals, with vertical bins spanning 8 m. Binned data were processed with the Cascade (Le Bot et al., 2011) software package provided by IFRE-MER. The Cascade procedure corrects for, among other quantities, the horizontal SADCP/navigation misalignment as well as misalignment of the horizontal plane due to ship roll. In addition, within Cascade the barotropic tidal component was removed

using the TOPEX/POSEIDON inverse model (Egbert et al., 1994).

## 2.3 Remote sensing data

Satellite-derived data (eg u,v, sea surface height, and surface chl-*a*) from altimetry and ocean color data were produced specifically for OUTPACE in the WTSP by Ssalto/Duacs and CLS with support from CNES. The procedures used to generate the data are similar to those described in (d'Ovidio et al., 2015), with the result that these products provide higher resolution and

quality control measures than is typically available from global products. As described in the following paragraphs, some of these products were produced post-cruise in delayed-time, while others were provided during the cruise in near real-time.

Maps of altimetry were generated delayed-time by merging along-track observations from the Jason-2, Saral-AltiKa, Cryosat-2, and HY-2A missions. The regional OUTPACE domain spans 140° E to 220° E, and 30° S to the equator, over the yearlong period of June 2014 to May 2015. The regional product has a resolution of $\frac{1}{8}^{\circ}$ using the FES2014 tidal model, and the

CNES_CLS_2015 mean sea surface. In order to produce this increase in resolution, among other measures region-specific noise measurements and correlation scales were calculated. Beyond the determination of merged absolute dynamic topography and its currents, corrections for cyclogeostrophy and Ekman effects at both the surface and 15 m depth have been included in separate products. Comparisons between Surface Velocity Program drifters deployed during OUTPACE and numerical Lagrangian particle experiments using these different products indicated that inclusion of an Ekman velocity component at 15 m produced a more accurate trajectory (Rousselet et al., this issue). Therefore, the Ekman-inclusive altimetry product was used

in the present study.

Both sea surface temperature (SST) and surface chl-*a* composite maps were also generated by CLS/CNES. These products were generated in near real-time during the OUTPACE campaign. Each map, with a $\frac{1}{50}^{\circ}$ resolution, uses a 5 day weighted mean of Suomi/NPP/Viirs measurements. The dataset spans from December 2014 to early May 2015.



## 2.4 Dynamic diagnostics and tools

The determination of the dynamical character of in situ currents surrounding the LDB bloom was implemented with use of the Richardson ($Ri$) number parameter. Classically, the gradient $Ri$ number is defined as:

$$Ri = \frac{N^2}{\left(\frac{\partial \boldsymbol{u}}{\partial z}\right)^2} \tag{1}$$

with $N^2$, the stratification, being the square of the Brunt-Väisälä frequency, and $\frac{\partial \boldsymbol{u}}{\partial z}$ being the vertical shear of horizontal velocity. $Ri$ is useful in characterizing differing regimes of flow, for example, the classical $Ri < \frac{1}{4}$ Kelvin-Helmholtz condition for instability. In submesoscale flows, instabilities such as Symmetric Instability (SI) appear when $Ri \leq 1$ (Stone, 1970). In order to determine how "submesoscale" the observed velocity shear is, we used a formulation from Thomas et al. (2013) to find the geostrophic component of shear, expressed as:

$$Ri_g = \frac{f^2 N^2}{|\nabla_h b|^2} \tag{2}$$

where $b = -\frac{g\rho}{\rho_0}$ is the buoyancy (with $g$ the gravitational constant, $\rho$ the density, and $\rho_0$ a reference density), $f$ is the Coriolis parameter, and $\nabla_h$ is the horizontal components of the gradient operator. In this paper, we characterized the flow as submesoscale when $Ri_g$ reached this value of 1.

The determination of $Ri$ required combining the MVP data for the numerator and SADCP data for the denominator, and so

is limited in resolution by the SADCP bins. Therefore, in calculating $N^2$, 8 m bins were produced by averaging the central finite differences from the 1 m density profile. Since we wished to characterize submesoscale phenomena, the horizontal gradient required for Eq.2 was left as is, without further re-binning. The calculation of $Ri_g$ was also calculated at the same 8 m resolution.

By assuming balanced flow, we can estimate the horizontal scale of density structures in a layer with the first internal Rossby

radius of deformation, approximated by Pedlosky (2013) as:

$$R_D = \frac{NH}{\pi f} \tag{3}$$

where $H$ is the depth of the water column (for non-constant stratification, this becomes an integral over depth). As a result, apart from latitude and depth, stratification plays a major role in determining at which horizontal scales one would expect to see balanced motions. Typically, the entire water column is used in this calculation, though, for order-of-magnitude approxima-

tions, other depths may be used. For example, a 5000 m CTD cast near station LDC during OUTPACE (not shown) provided a radius of 65.2 km, consistent with a global climatological atlas for this region (Chelton et al., 1998). The average stratification during LDB produced scales of 160 m, 570 m, 2.7 km, 5.8 km, and 9.7 km for the top 20, 30, 50, 100, and 200 m, respectively. Using these scales, the horizontal resolution of the MVP transects sufficiently resolved features affecting layers comprising the upper 50 m and deeper.

The mesoscale structuring of the chl-$a$ bloom was evaluated based on altimetry-derived finite size Lyapunov exponents (FSLE), computed following the algorithm of d'Ovidio et al. (2004). FSLEs were calculated by time-integrating trajectories on





a grid using the velocities described in Sect. 2.3.1. The trajectories were found with a fourth-order Runge-Kutta method with a 6 h timestep. Particles were initially separated by 0.05° (the grid resolution) and reached a final separation of 0.6°. Velocity fields were linearly interpolated in space and time.

All FSLEs were calculated using 30 day backward integrations, meaning that the 'final' separation reflects initial displace-
ments that were subsequently brought together (ie convergence). FSLE maximal values often form lines, or ridges, that were used to identify possible frontal zones of enhanced strain, along which tracer gradients should generally align. The robustness of FSLE calculations to small-scale errors in velocity fields has been previously studied (Cotté et al., 2011). Additionally, in practice FSLEs have been shown to be a useful heuristic in identifying flow manifolds in two-dimensional data. In effect, the FSLEs here represent the emergent structures one would expect to see if the larger-scale mesoscale features visible by satellite
were important.

The altimetry-derived currents produced by CLS/CNES were also used to calculate Lagrangian particle trajectories both backward and forward in time, through the use of the Lagrangian diagnostic tool ARIANE (Blanke and Raynaud, 1997; Blanke et al., 1999; Blanke, 1997). Generally, this tool uses either model or empirical output to drive particle trajectories. Since the dominant tracer in this study, chl-*a*, is reactive, the initial point for integration of trajectories for ARIANE was somewhat
subjective. As a result, we chose here to start when the bloom associated with LDB was at its pinnacle point, namely 6 March, 2015 as judged by satellite data. It is reasonable to assume that seeding Lagrangian particles from this maximal point in the surface bloom gave the best chance that the particles would remain in the sphere of bloom water, and would point to both the potential source of the bloom and its endpoint. Lagrangian particles were spaced $\frac{1}{50}°$ ($\sim 2$ km) apart within the chl-*a* contour of 0.3 mg m$^{-3}$.

# 3  Results

## 3.1  Chl-*a* and density distribution and timeseries

### 3.1.1  MVP Observations

The four MVP transects conducted before and after LDB are shown in Fig. 2. The distributions of chl-*a* fluorescence and isopycnals are presented in panels Fig. 2b-e. During T1 (Fig. 2b), the chl-*a* maxima was initially located near 120 m depth,
where it was then present at 100 m before being found at the surface near 77 km, inside the patch. Within the patch, a surface signal was present from the surface to 40 m, with a diminished DCM at 80 m. Outside the patch, at 127 km, the DCM was again found near 120 m.

T2 (Fig. 2c) demonstrated a similar pattern, with the DCM found from 120 to 100 m again, reaching the patch edges at 110 and 222 km. The DCM within the patch was less distinct from the surface signal, while the strongest chl-*a* values were
still located within the top 40 m. T3 (Fig. 2d), which followed the same route as T2 but occurred some hours later and in the opposite direction, showed the same structure: a 40 m surface patch located at 22 and 123 km that was not distinct from the DCM, with the DCM found again at 100 and then 130 m by the end of T3, outside the bloom.





The distribution in T4 (Fig. 2e) was sampled a week after T1-T3 and subsequent to station LDB. The bloom was present from 17 to 110 km along T4. In this case, a separate DCM at 70 m was present below the surface signal. Unlike previous transects, the DCM dropped abruptly to 140 m at 90 km, well before the end of the surface bloom at 110 km.

MLD and density contours throughout these transects were relatively flat, with the MLD centered around 19 m and the 22, 23, 24, and 25 density anomaly contours centered at 16, 41, 83, and 182 m, respectively. Among these isopycnals, only the 22 contours reached above the first depth detected by the MVP at 10 m. All of these outcroppings occurred either within the bloom area or adjacent, with the exception of the last at the end of T4.

### 3.1.2 CTD timeseries

The chl-*a* structure of LDB, as observed during the 47 CTD casts, is shown in Fig. 3. From the beginning until 18 March, the maximum value of chl-*a* could be found between the 22 and 23 density anomalies, around 30-40 m. After 18 March, the chl-*a* max was found at a deeper position near 60 m. Over time, the chl-*a* max deepened, and was found between the density anomaly contours of 23.5 and 24, or 60-80 m depth. From 18 March onward, the surface concentration of chl-*a* also decreased, whereas the chl-*a* max concentration increased and began to resemble a typical DCM distribution.

Throughout LDB, the density anomaly contours remained relatively flat. The 22 isopycnal did not outcrop near the surface, unlike that seen in the MVP transects, though the position of LDB within T2 and T3 was inside the region where the 22 isopycnal was found at a similar depth. Upward and downward oscillations of isopycnals indicated the presence of near-inertial oscillations, which was reflected in the SADCP timeseries that is investigated further in (de Verneil et al., this issue). The MLD, though not formally tied to an isopycnal value, also largely reflected these oscillations. Nitracline depth, which in our formulation was tied to the 24.34 isopycnal, was consistently found between 100 and 120 m throughout the dataset, well below the chl-*a* max and general bloom area, even when the DCM distribution began to appear towards the end of LDB.

### 3.2 Physical and biogeochemical water properties

Observations of SD12, LDB, and SD13's water structure and biogeochemical properties are presented in Fig. 4. The three stations had similar temperature and salinity profiles, as indicated in the T-S plot of the upper 200 m (Fig. 4a). All three stations covered a similar density range, with slight variation in their salinities. At the surface, SD12 had slightly lower salinity, while LDB and SD13 overlapped more. At depth, however, SD12 and LDB had closer salinity values, with SD13 showing a salinity maximum at depth.

The nitrate profiles of all three stations (Fig. 4b) showed concentrations near the quantitative threshold limit of the measurement method above 100 m. SD12's nitrate increased the fastest, starting at 100 m whereas LDB and SD13 matched each other, both increasing at 120 m. The different depths at which nitrate began to increase are similar to the DCM present at both SD12 and SD13 (Fig. 4c), with the DCM near 100 and 120 m, respectively. The chl-*a* profiles for the beginning and end of LDB reflected the changes shown in Fig. 3. Phosphate concentrations (Fig. 4d) for all three stations were nearly equal at 85 m depth, and generally increased with depth. Above 85 m however, the concentration decreased for SD12 approaching the surface, though it was always above the quantitative measurement threshold. LDB demonstrated a similar decline in phosphate





near the surface, but the values were consistently below the detection limit in the top 40 m. By contrast, phosphate values for SD13 were near a steady concentration up to the surface.

### 3.3   In situ $Ri$ and $Ri_g$

The possible presence of balanced, submesoscale currents in the in situ MVP and SADCP datasets was determined by compar-

ing $Ri$ and $Ri_g$. Since both $Ri$ and $Ri_g$ span several orders of magnitude, $\log_{10}Ri$ and $\log_{10}Ri_g$ during T4 are shown in Fig. 5. All the transects contained similar distributions, and T4 was chosen because of the sharp gradients in chl-$a$. The majority of observations of $\log_{10}Ri$ and $\log_{10}Ri_g$ were above 0, or 1 in normal space, and so here we diagnosed that the circulation was largely not submesoscale in its dynamics. $\text{Log}_{10}Ri$ (Fig. 5a) values were similar to but slightly lower than $\log_{10}Ri_g$ (Fig. 5b), demonstrated by a preponderance of saturated shading in 5b. Larger $\log_{10}Ri_g$ values were found at depth, with the smallest

$\log_{10}Ri_g$ found near the surface. In the top 50 m, the mean of $\log_{10}Ri_g$ was 1.38, whereas it was 1.54 for the entire profile. By contrast, $\log_{10}Ri$ was more consistent between the top 50 m and all depths, with a mean of 1.35 and 1.36, respectively. Only 2.2% of $\log_{10}Ri_g$ was less than or equal to 0, with 6% for $\log_{10}Ri$. Additionally, for values less than or equal to 1, 39% and 35% of $\log_{10}Ri_g$ and $\log_{10}Ri$, respectively, reached these levels for all depths. $Ri$ reached a minimum near the surface for the last 40 km of T4, which was not reflected in $Ri_g$. Some structures in $Ri$ appeared in a diagonal orientation, such as between

75 and 125 km at 100 m depth. Vertical patterns appeared in $Ri_g$, spanning approximately 10 km horizontally, where several isopycnals were concurrently found closer to the surface or deeper. One clear example of this occurred near 25 km within the upper 50 m. Beyond the minimum $Ri$ values at the surface near the end of T4, there were no distinguishing features for either $Ri$ or $Ri_g$ between being inside the chl-$a$ bloom and outside, neither at the surface nor at depth.

### 3.4   Satellite chl-$a$ timeseries, altimetry-derived FSLE, and ARIANE trajectories

The remotely sensed distribution of surface chl-$a$, calculated FSLEs and ARIANE Lagrangian particle positions, over a period spanning 25 December, 2014 to 10 May, 2015 are shown in Fig. 6. FSLEs and particles are shaded gray and red, respectively, with 10 % of the particles randomly selected for plotting in all subpanels. 25 December (Dec., all months hereafter shortened) was chosen as the starting point by visually examining the chl-$a$ dataset for a pre-bloom period, with a bloom 'source' region identified on 13 Jan. centered at 186° E, 20° S.

The 25 Dec. start date showed a modest chl-$a$ region oriented North-South (N-S) near 186° E (Fig. 6a) with several likewise N-S oriented FSLEs both inside the chl-$a$ region and to the east. Lagrangian particles were dispersed over the northern half of the defined region of interest. On 13 Jan., the identified source region showed the chl-$a$ patch localized near an island group located at 20° S, 186° E (Fig. 6b). Multiple FSLE ridges were stacked near the chl-$a$ patch, indicating the likely flow along these trajectories to the east. Additionally, another ridge appeared, oriented N-S near 189.5° E, and aligned with a weaker chl-$a$

gradient. The Lagrangian particles had now advected west and were flowing south, near the island group 'source' region.

On 31 Jan. (Fig. 6c), the FSLEs to the south of the source island group had evolved into a recirculation pattern to the South, creating a lobe of low chl-$a$ water near 20.5° S, 187° E. The ridge near 189.5° E on 13 Jan. had moved westward, and collided with the eastward-flowing chl-$a$, creating a N-S stretching of the bloom. The northward flowing arm of the bloom now





approached an East-West (E-W) FSLE ridge (17°S, 187° to 189° E). At this time, the particles had likewise begun to move from the island group to the east, with some particles overlapping the high chl-*a* bloom.

By 16 Feb. (Fig. 6d), the E-W FSLE ridge was now slanted, as was the chl-*a* patch's northern boundary. To the south, the FSLE ridge that collided with the patch on 31 Jan. had begun to move west, as had the chl-*a* patch spanning 22° to 20° S, 187°
E. High chl-*a* concentrations positioned at 18.5° S, 189° E had advected eastward between the gap in the E-W FSLE ridge and the N-S ridge that collided with the bloom near 31 Jan. The Lagrangian particles were mostly within the elevated chl-*a* region, and like the bloom water they largely did not cross the FLSE ridge.

On 6 Mar. (Fig. 6e and f), the E-W FSLE ridge had stabilized the formerly northward flowing bloom waters, and another N-S ridge near 191° E began to move in from the east. Another FSLE ridge embedded within the southern bloom lobe positioned
near 20° S, 187° W, indicated possible strong flow inside the bloom itself. More E-W ridges at 21° S, 188°, E also appeared near the bloom's southern boundary. Farther south, apart from the current bloom of study, bands of high and low chl-*a* water appeared amidst E-W FSLE structures. These structures were more or less present from this point on until the end of the study period. Lagrangian particles for 6 Mar. were all located within the high chl-*a* region since this was the chosen particle initialization time.

By 21 Mar. (Fig. 6g), the day of MVP T4 sampling, the N-S ridge from earlier had collided with bloom waters and stopped their eastward transit (18° S, 191° E). The circulation inside the bloom, coinciding with a ridge from 20° to 18° S, 188° E, had entrained low chl-*a* water near the former center of the bloom, producing two lobes of high chl-*a* regions to either side. The particle distribution likewise had these two lobes, and the southern particles had begun to move west, beginning the shearing apart of the chl-*a* bloom.

3 Apr. (Fig. 6h) began to show the general decrease in chl-*a*, a trend that continued until 24 Apr. and 10 May (Fig. 6i-j). The N-S FSLE values near T4 on 21 Mar. (now 18° to 19° S, 189° E) had advected westward, along with the recirculation ridge (20° to 17° S, 187.5° E) and the remnants of the bloom at LDB. Particles had now spread apart, largely overlapping the decreasing chl-*a* bloom and oriented along FSLE ridgelines. Another E-W ridge (17.5° S, 189°-190° E) asserted itself by 24 Apr., oriented along a new post-bloom boundary between moderate and low chl-*a* values. 10 May, the end of this study period,
now had a region of minimum chl-*a* in the exact region where the bloom was near its peak chl-*a* satellite values on 6 Mar. (compare Fig. 6f and j). Numerous N-S and E-W ridges, generally aligned with the remnant chl-*a* gradients, continued to move west, along with particles generally embedded within them.

Statistical properties of the Lagrangian particle positions and their respective chl-*a* values are presented in Fig. 7. The percentage of Lagrangian particles that stayed within the study region is shown in Fig. 7a. Throughout the entire period of
consideration, the percentage of particles within the region was always greater than 70%. During the backward integration, over 95% of the particles were present until 13 Jan., the date when the bloom was localized near the island group. Prior to this date, the percentage dropped until it was near 70% on the first day, 25 Dec., 2014. In the forward integration, the proportion of particles steadily dropped, until it was around 80% in early to mid-April, after which it rose. The downward trend slowly reasserted itself, and by 10 May, the end of the period, the percentage had dropped to near 70% like in the backward integration.



The mean value of chl-*a* at the particle positions is depicted along with the mean satellite value, as well as the 1, 25, 75, and 99% satellite values in Fig. 7b. Moving in either direction from the 6 Mar. starting point, the mean particle chl-*a* concentration dropped from its initialization peak. The particle mean chl-*a* value was consistently above the mean satellite value, except for a short period at the end of the backward integration in Dec. 2014. Particle mean values reached the 75% threshold 16 Feb. and

3 Apr. in the backward and forward integrations, respectively. The satellite 99% value rose from Dec. 2014 to 13 Jan., when the bloom was identified near the island group. The 99% value varied around this point until reaching its maximum shortly after 6 Mar., the initialization date for the particle experiments. After this point, the 99% satellite value began to decline, with a precipitous drop after 21 Mar., the date of MVP T4.

## 4   Discussion

### 4.1   Chl-*a* bloom processes, physical forcing, and collapse

The bloom sampled in this study had a number of distinguishing features that differentiate it from the surrounding water sampled during OUTPACE. Together with physical measurements, these characteristics allow for a determination of what combination of biogeochemical processes and physical forcing were responsible for the bloom and its temporal evolution.

Since this study focuses on a surface bloom, there must be some process responsible for the relative accumulation of chl-*a*.

Ideally, in situ biological rate measurements would have been measured before and during the chl-*a* increase. In addition to the pragmatic difficulty of obtaining all these measurements for all time and space, the satellite data suggest the bloom was already two months old by the time of sampling at station LDB, precluding direct observation of its initial conditions. Therefore, one must start with the assumption that the starting conditions for the bloom were similar to surrounding regions (eg the chl-*a* distribution pre-bloom on 25 Dec., Fig. 6a). We can infer, then, that the local increase of chl-*a* in the bloom was linked to a

source of new production. For the WTSP, the two main mechanisms to consider are first nitrate delivery from below due to advective fluxes and/or diapycnal mixing, and secondly $N_2$ fixation by diazotrophs.

The data from stations SD12, LDB, and SD13, support the role of $N_2$ fixation in creating the bloom and reject nitrate delivery from below. Firstly, the high chl-*a* concentrations in the bloom were located in the top 40 m, as shown by the in situ MVP surveys (Fig. 2) and station LDB CTD timeseries (Fig. 3). The nitracline, one potential source for new production, was

consistently at least 60 m deeper in the water column in the CTD timeseries than the bloom. In order to provide an advective flux, the isopycnals present would have at some point needed to traverse the nitracline depth to the bloom at the surface. This did not occur anywhere within the CTD timeseries. Additionally, the only isopycnal that outcropped during the MVP transects passing inside and outside of the bloom, the 22 kg m$^{-3}$ anomaly, normally resided in the upper 30 m. Therefore, the stable stratification and lack of horizontal density gradients spanning the top 100 m rule out the potential for a vertical advective

flux of nitrate to the surface bloom. These observations do not preclude the possibility that an initial nitrate flux occurred two months prior during the bloom's inception, with the water column subsequently mixing and re-stratifying. This possibility is highly unlikely, considering that stations SD12 and SD13 had similar T-S characteristics and nitrate profiles to LDB, and yet




had a DCM instead of a surface chl-*a* bloom (Fig. 4). These data therefore also remove the possibility of a massive diapycnal mixing event.

The lack of evidence for mixing events or advective nutrient fluxes leads us to consider the parsimonious alternative that the bloom was supported by $N_2$ fixation as the source of new production. Diazotrophs, the organisms responsible for $N_2$ fixation, are normally concentrated in the surface layer, exactly where the bloom was found, and during station LDB this process was observed directly (Caffin et al., this issue). Evidence for $N_2$ fixation's role was also reflected in the nutrient profiles. Nitrate levels in the top 100 m were consistently below the quantitative threshold, unsurprising since nitrogen is normally the limiting nutrient. Once nitrogen fixation relieves nitrogen limitation at the surface, other inorganic nutrients such as phosphate are consumed. The decreasing concentration of phosphate near the surface at station LDB, as opposed to the constant values at station SD13, supports this interpretation. Interestingly, station SD12 showed a similar decrease, yet not as strongly. This allows for the possibility that stations SD12 and LDB were more alike in the fact that at some point $N_2$ fixation was occurring in situ, though the DCM structure at SD12 and SD13 resembled each other more. The possibility that SD12 also hosted nitrogen fixation, whereas SD13 did not, fits within the overall, regional gradient in nitrogen fixation that OUTPACE set out to observe going toward the SPG. For the purpose of this study, it is sufficient to note that the extent of nitrogen fixation present at LDB was likely much larger than at SD12 and in surrounding waters, considering the observed phosphate depletion and the long-lived surface chl-*a* signal associated with the bloom.

Station LDB's CTD timeseries also showed the decrease of surface chl-*a* and the new formation of a DCM near 80 m. MVP T4 likewise showed the concurrent surface bloom and the appearance of a DCM at 70 m immediately after station LDB. As previously mentioned when considering nutrient delivery from below, the relatively flat isopycnals also precluded the reverse movement, namely physical subduction of chl-*a* away from the surface. Instead, other mechanisms must be considered. The passive movement of chl-*a* from the surface to a DCM through sinking was possible, but not likely to be a major sink due to the small cell sizes of the plankton concerned (though TEP may be a factor; (Berman-Frank et al., this issue)). In situ removal of chl-*a* may have occurred for a number of reasons. First, the exhaustion of phosphate at LDB may have led to a decline in production, which could not match the removal processes of senescence and grazing. Indeed, dissolved inorganic phosphate (DIP) turnover times reached 0.1-0.2 days above 40 m depth, a value largely below the critical value of 2 days necessary for *Trichodesmium* spp., a major $N_2$ fixer in this area, to grow (Moutin et al., 2005). Second, the bloom may have fallen victim to increased predation, as abundance of some zooplankton was enhanced (Dolan et al., 2016). Interestingly, even though these biological mechanisms are mediated in situ by planktonic organisms, the regional and simultaneous collapse of the bloom was documented by satellite imagery (Fig. 6 g-h), meaning the same mechanisms most likely acted throughout the bloom. The chl-*a* structure at the end of station LDB resembled that of SD12 much more than its initial distribution at the beginning. This observation, in conjunction with evidence of nitrogen fixation near the surface, raises the possibility that at some point station SD12 underwent a similar process, and that surface blooms may play a more general role in the region.



## 4.2 Surface circulation, chl-*a* advection, and the formation of gradients

The major focus of this study, apart from diagnosing the bloom's biogeochemical makeup, is to identify which circulation patterns explained the horizontal distribution of the bloom once it was formed, namely submesoscale or mesoscale currents. Comparing the in situ datasets with satellite altimetry currents, the relevant scales of surface motion in the WTSP can be deduced.

As mentioned in the introduction, though the SEC is the mean current advecting water westward from the northern limb of the subtropical gyre, much more kinetic energy is typically found at the mesoscale. Interest has mounted in recent decades in the possibility that submesoscale currents, intensified at the surface, may play a role in affecting nutrient delivery into the euphotic zone, as well as redistributing biological tracers, including in $N_2$ fixation zones (Calil and Richards, 2010). While nutrient delivery has been ruled out for this particular bloom, there remains the possibility that submesoscale motions played a role in the lateral advection of the surface bloom, as well as providing the shear necessary to form the strong chl-*a* gradients observed in the MVP transects. In particular, recent findings highlight both the seasonal (Callies et al., 2015) and geographic (Callies and Ferrari, 2013) occurrence of submesoscale turbulence; for the WTSP, it is unclear whether it should be favored or not. On one hand, it is located away from regions of strong baroclinic currents such as western boundary currents, and according to eq. (2) the reduced Coriolis parameter relative to temperate latitudes should slightly reduce the geostrophic $Ri_g$. Conversely, the late summer stratification may inhibit these motions, making the end result of these competing effects difficult to predict a priori.

The values of $Ri$ and $Ri_g$ present within the top 50 m of the MVP transects were predominantly greater than 1, with the majority larger than 10 (0 and 1 in log-space, respectively). Considering these two independent measurements of shear agreed in magnitude, and that both occurred outside the dynamical range of consideration, the data thus provided little support for the presence of submesoscale circulation, perfectly balanced or otherwise.

Combining the strong surface stratification, general lack of isopycnal outcroppings (ie no strong horizontal density gradients), and absence of correspondence between chl-*a* and density gradients, it should perhaps not be surprising that buoyancy-driven submesoscale circulation was not readily discernable even at the surface in this dataset. If most submesoscale structures are expected to be in the mixed layer, which throughout this dataset was near 20 m, then the sub-kilometer Rossby radius $R_D$ (here found to be $< 200$ m) would not have been resolved by the MVP survey. However, since the bloom of interest in these surveys spanned the top 40 m, and covered hundreds of kilometers in horizontal extent, these small features, should they have existed, would not have impacted the full depth range of the bloom, nor would they have significantly affected the horizontal advection of the entire bloom. Furthermore, the strongest gradient in chl-*a* found in MVP T4 had no visible density structure at these scales to suggest a source of horizontal shear to create this gradient.

The possibility remains that during other periods of the year such as winter, with a deeper mixed layer than that observed during the cruise, the horizontal spatial scales of these features would increase and possibly impact the advection of such a bloom. The question thus becomes whether these bloom events occur during winter periods. If not, then the mutually exclu-





sive timing of bloom events and stratification favorable for submesoscale turbulence would preclude its influence on bloom evolution for the region.

The natural horizontal scales of the bloom were more in line with the $R_D$ of the entire water column, $\sim 60$ km, which is impacted by the mesoscale regime. In our dataset, satellite altimetry-derived currents represented the mesoscale circulation.

Whereas the MVP T4 transect revealed no submesoscale source of horizontal shear to create a chl-$a$ gradient, the mesoscale circulation did via an FLSE structure present near MVP T4 in Fig. 6g. While FSLEs may not provide the exact regions where chl-$a$ gradients form in the station LDB bloom, they help define dynamical boundaries that provide more explanation than the in situ data for why the chl-$a$ bloom appeared as it did.

The positions of Lagrangian particles also allow for a better representation of where bloom water advected, as shown in

Fig. 7. Firstly, most particles were present in the defined region of interest throughout the timeseries (Fig. 7a), providing additional support (beyond visual inspection) that the chl-$a$ dataset sufficiently covered bloom water relative to the chosen initial time of integration, March 6. If the chl-$a$ satellite timeseries therefore encompasses most of the bloom water of interest, then the fraction of particles in high chl-$a$ water throughout their trajectories, as represented in Fig. 7b, is an indication of the performance of these particles in keeping with the bloom. Evidently, chl-$a$ is a reactive tracer and undergoes its own evolution,

as shown by the shaded areas in Fig. 7b. The ability of the Lagrangian particles to both remain in the region of interest and to accurately represent elevated chl-$a$ values (mostly above or near the 75th percentile) provides strong positive evidence that mesoscale flows were indeed advecting the bloom water.

With accurate representation of the advection of bloom water, an interesting picture emerges. Firstly, the passage of particles near an island group before and at the beginning of the bloom (Fig. 6a-b) suggests a possible island effect in the ignition of

the bloom. Considering that $N_2$ fixation drove new production, and nearby stations SD12 and SD13 had detectable phosphate levels, alleviation of another necessary and limiting nutrient, iron, was possibly at work. The enrichment of diazotrophs near island inputs, and their subsequent advection (along with their primary production), has been documented previously (Dupouy et al., 2013; Shiozaki et al., 2013). Secondly, the shifting FSLEs and Lagrangian particle tracks demonstrate the general eastward advection of the bloom from its localized island source in Fig. 6b until its easternmost position in Fig. 6g. This

eastward evolution is not what one would expect a priori. Indeed, both the SEC, as well as the mesoscale structures responsible for currents in the altimetry dataset, propagate westward, reflected by the migration of N-S oriented FSLE ridges the bloom encountered in Fig. 6b-d and Fig. 6f-g.

Therefore, the complex circulation evoked by a westward-moving mesoscale field allowed for the counterintuitive eastward advection of water with enhanced biological production at its surface. This was possible due to the bloom occurring in water not

associated with the coherent, elliptic structures that move west, and which are what most investigators focus on for mesoscale transport. Instead, the bloom occurred in water outside these structures, with tortuous trajectories hyperbolic in nature (Kirwan et al., 2003). Eventually, as the bloom collapses, more particles move west with the mean circulation, as shown in Fig. 6h-j. The temporal overlap between the surface bloom's occurrence and maintenance until its collapse, in conjunction with its easternmost transport, is possibly a coincidence. It does suggest, however, the important role that complex mesoscale flows

outside of coherent, elliptic eddies have in determining where new production eventually ends up.





## 5 Conclusions

In this study, we document a surface chl-*a* bloom observed in the WTSP spanning two months from mid-January until mid-March 2015. Large-scale in situ surveys conducted by an MVP platform both confirm the surface signal seen by satellite and further show the lack of the DCM common in the region. A quasi-Lagrangian CTD timeseries additionally shows the collapse
of the surface bloom, also corroborated by satellite. Through the use of chl-*a*, density, and nutrient profiles, the delivery of nutrients from depth is ruled out as a mechanism sustaining the bloom. Instead, the surface bloom hosted significant $N_2$ fixation as a source of new production. In the WTSP, the hotspot for $N_2$ fixation in the world, the surface inorganic phosphate from the South Pacific gyre may represent an important nutrient source for maintaining such blooms subsequent to sufficient introduction of iron, possibly due to island effects.

The circulation responsible for advecting the bloom at the surface was satisfactorily represented by altimetry-derived mesoscale currents, with physical structures corresponding to biological gradients. The in situ density and velocity data, in contrast, did not have structures that correspond with the sharp gradients in chl-*a*. Additionally, the dynamical nature of the in situ data did not fall within the submesoscale regime. In the ongoing debate between the seasonal and geographic distribution of submesoscale turbulence, this dataset posits the first-order importance of mesoscale circulation in the WTSP summer.
The complex trajectories that mesoscale currents create in hyperbolic regions outside of eddies can be of prime importance in advecting blooms, sometimes in counterintuitive directions such as the eastward transport presented here.

Future studies in the region will be necessary to resolve residing questions. For instance, the particular biogeochemical conditions igniting the bloom, here hypothesized to be island-derived iron, cannot be confirmed within this dataset and will require further in situ sampling. Additionally, the presence of submesoscale structures during other parts of the year, such as
wintertime with deeper mixed layers, remains unanswered.

However, armed with the information that not only summertime blooms fueled by $N_2$ fixation occur in the WTSP, but that their advection at the surface may be represented by the value-added, high-resolution satellite products such as those produced by CLS/CNES, future studies may begin to quantify the combined inter-annual impact of $N_2$ fixation and mesoscale transport in distributing new production in this region.

*Acknowledgements.* This is a contribution of the OUTPACE (Oligotrophy from Ultra-oligoTrophy PACific Experiment) project (Moutin and Bonnet, 2015) funded by the French national research agency (ANR-14-CE01-0007-01), the LEFE-CyBER program (CNRS-INSU), the GOPS program (IRD) and CNES (BC T23, ZBC 4500048836). The OUTPACE cruise was managed by MIO (OSU Institut Pytheas, AMU) from Marseilles (France). The authors thank the crew of the RV *L'Atalante* for outstanding shipboard operations. G. Rougier, M. Picheral, and L. Bellomo are warmly thanked for their help in CTD rosette and MVP deployment and data processing. We also thank A. Gimenez for
$T_{DIP}$ measurements, and S. Helias-Nunige for nutrient values. C. Schmechtig is thanked for LEFE-CyBER database management. Satellite SST, Chl-*a*, and altimetry data have been provided by CLS in the framework of CNES funding; we thank I. Pujol and G. Taburet for their support in providing these data. A. Lozingot is acknowledged for administrative aid for the OUTPACE project.



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





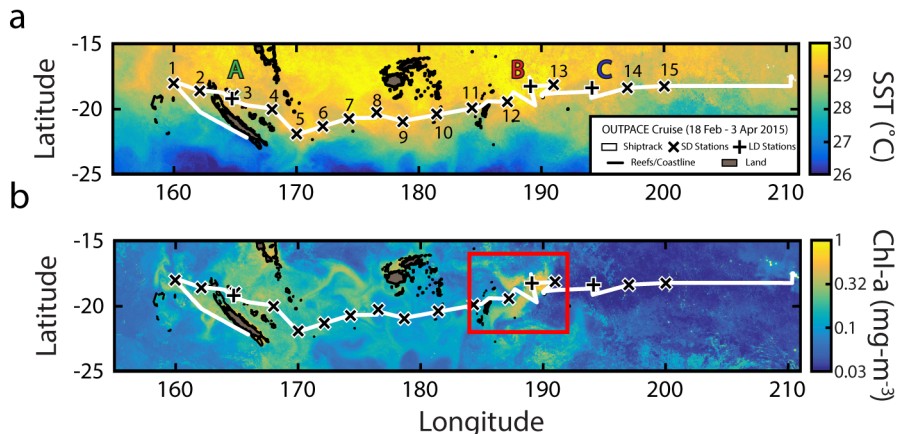

**Figure 1.** OUTPACE satellite (a) SST and (b) Chl-*a*. Satellite pixel data over 42 days are used to produce a weighted mean. The weight for each pixel is calculated by normalized inverse distance squared from the pixel to the daily mean ship position. The shiptrack, derived from ADCP data, is shown in white, land is shaded gray, with black coastlines and reefs. Short duration (SD) stations are depicted by black X marks, while long duration (LD) stations are shown by black +'s. The LD stations are color-coded, using the convention for the OUTPACE cruise. A red rectangle in the Chl-*a* panel shows the study area for this paper.





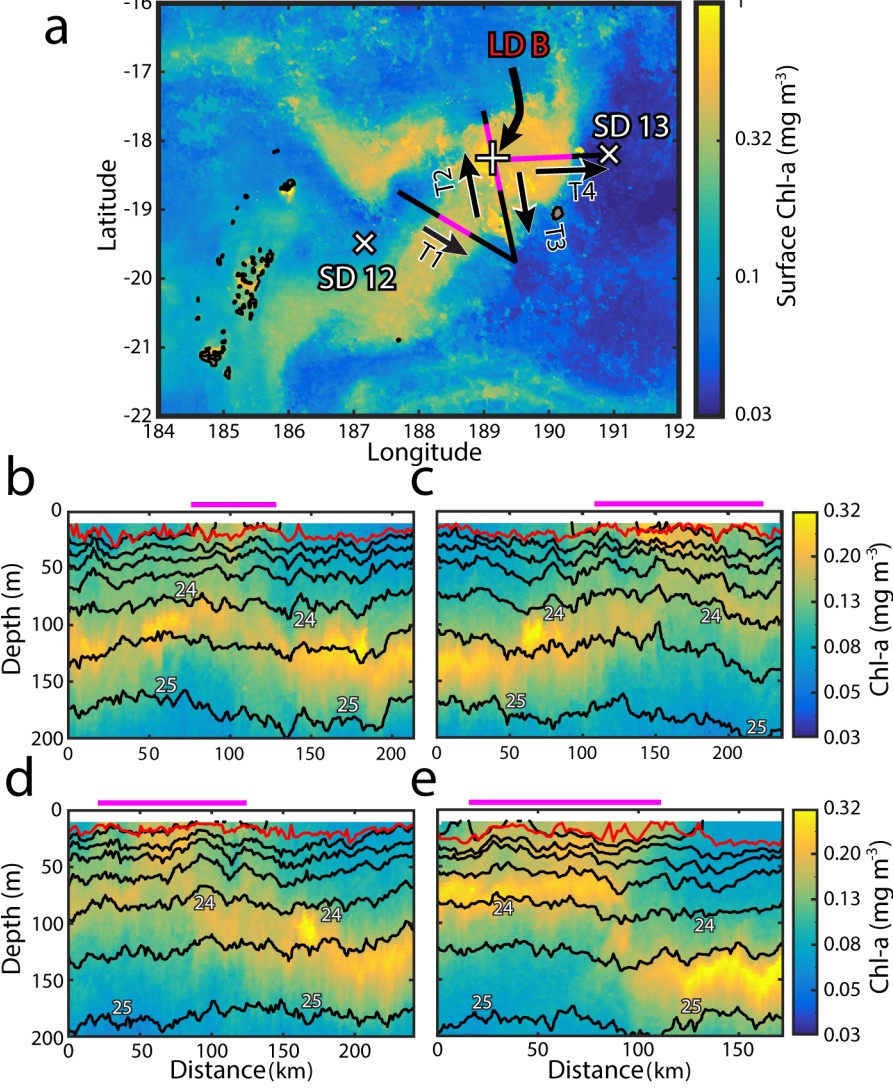

**Figure 2.** MVP Transects near the station B bloom site. (a) The position of each transect with respect to Chl-*a* fluorescence transects for (b) Transect 1 (T1), (c) Transect 2 (T2), (d) Transect 3 (T3), and (e) Transect 4 (T4). Transect positions are shown by black lines, with the casts designated inside the bloom by MVP data are colored magenta. Stations SD12, LDB, and SD13 are depicted by marks similar to Figure 1. Direction of sampling for the transects are shown by arrows. Islands and reefs are also shown in a similar fashion to Figure 1. In situ isopycnals are shown in black, ranging from 22 to 25 $kg-m^{-3}$ in half-steps. The MLD is shown in red, and the magenta brackets above the panels show the casts inside the bloom.





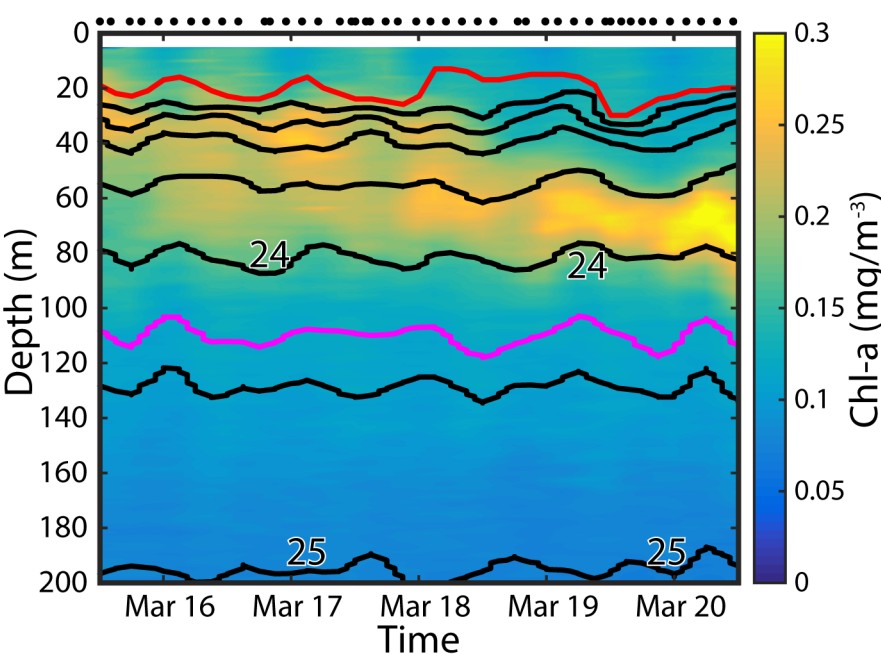

**Figure 3.** LDB CTD timeseries (in UTC) of Chl-*a*. Contours of density anomaly are superimposed in black ranging from 22 to 25 kg m$^{-3}$ in increments of 0.5. MLD is shown in red, with $D_{NO3}$ in magenta. Black circles above the panel indicate times of CTD casts.





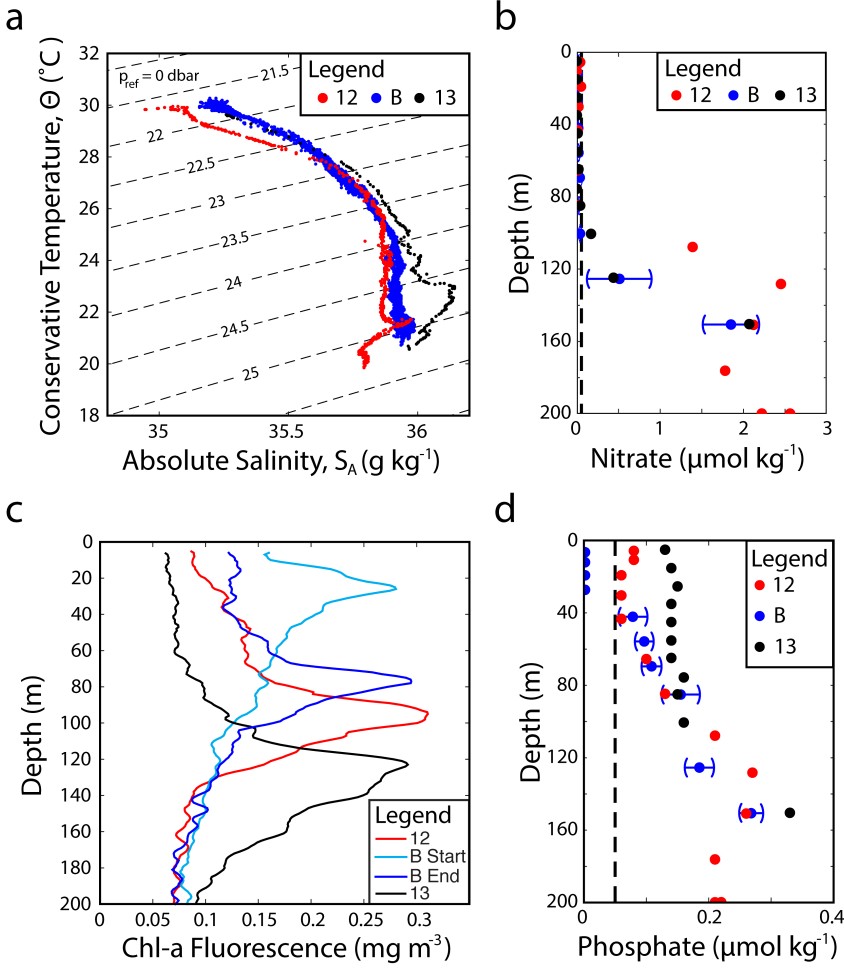

**Figure 4.** Station SD12, LDB, and SD13 CTD and bottle data. (a) $\Theta$-$S_A$ diagram, (b) $NO_3$, (c) Chl-$a$, and (d) $PO_4$ concentrations. SD12 data is shown in red, LDB in blue, and SD13 in black. Quantitative thresholds for $NO_3$ and $PO_4$ are shown by black dashed lines, and any observed values below this threshold are set to 0. The 95% confidence intervals are included for the LDB nutrient values.





**Figure 5.** Calculated (a) $Ri$ and (b) $Ri_g$ for MVP T4. Density contours are superimposed in black and MLD in red, with magenta indicating the region inside the bloom, similar to Figure 2e. Please note the $log_{10}$ color scale.





**Figure 6.** Satellite chl-*a*, FSLE, and ARIANE particles for (a) 25 Dec. 2014, (b) 13 Jan. 2015, (c) 31 Jan., (d) 16 Feb., (e-f) 6 Mar., before station LDB sampling, followed by (g) 21 Mar., the date of MVP T4, concluded by post-bloom (h) 3 Apr., (i) 24 Apr., and (j) 10 May. FSLE values above $0.15 \, \mathrm{day}^{-1}$ are shaded gray. A randomly selected subsampling of one-tenth of the ARIANE particles was initially chosen to be shown in red.





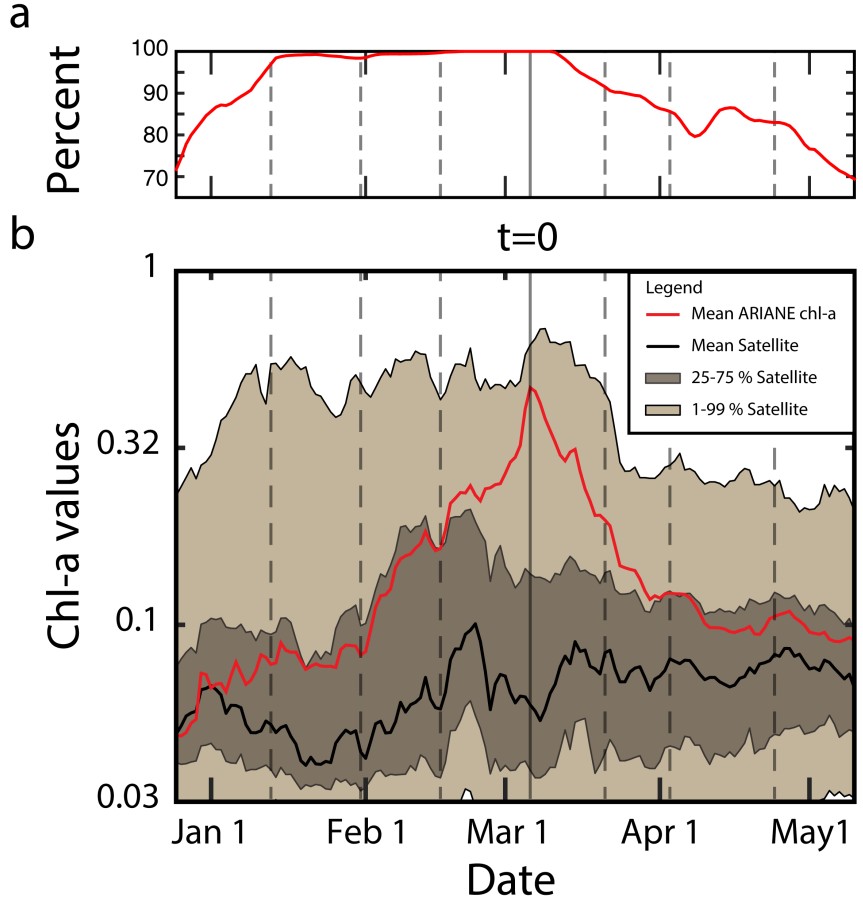

**Figure 7.** Statistics from ARIANE lagrangian particle trajectories, from Dec 24, 2014 to May 10, 2015, with initial time Mar 6 shown with a solid vertical gray bar, and the dates from figure 6 in dashed lines. (a) Percentage of particles found inside the bloom region, between 184° to 192° E and 22° to 16° S. (b) Mean interpolated chl-$a$ value for all lagrangian particles during backward and forward integrations in red, with mean chl-$a$ from the CLS dataset in black. Dark shading shows region between $25^{th}$ and $75^{th}$ percentiles, with light shading for the $1^{st}$ and $99^{th}$.