# Peer review of "The Fate of a Southwest Pacific Bloom: Gauging the impact of submesoscale vs. mesoscale circulation on biological gradients in the subtropics"

_Biogeosciences, 2017_

## Referee Comment (RC1) · Anonymous Referee #1 · 19 May 2017

The topic of how primary production may be sustained by mesoscale and submesoscale circulations, particularly in the low nutrient subtropical gyres, is one of considerable broad interest. Yet it is an area where observations remain relatively few, particularly compared to modelling studies despite several of the latter indicating a major role for such physical processes. Motivated by encountering an unexpected phytoplankton bloom in such a region, the authors use a combination of in situ data and remote sensing to explore the potential role of the physical circulation. Their conclusion - that vertical nutrient flux from submesoscale processes is an unlikely control and that it is largely a 2D phenomenon with the mesoscale field advecting water against the

mean flow – is one that seems best supported by the data. This is a very interesting result, particularly with the corollary that the mesoscale flow may be drawing iron away from the islands to trigger the bloom. However, there are a few aspects of the paper that I think need addressing before the manuscript can be published.

The paper essentially considers two possibilities: submesoscale vertical movement and mesoscale horizontal movement. The reader needs to have faith that both options have been thoroughly tested before the conclusion has been reached.

For the submesoscale, the largely horizontal structure of density and discontinuities between surface and deep Chl are pretty convincing. I find the Richardson number argument less so given that the majority of submesoscale motions are confined in the surface boundary layer which is poorly sampled judging by Fig. 5. The authors should acknowledge this. For clarity, Fig 5 should also have its colour scale changed so that the blue-yellow transition is centred on zero – the value of interest. Additionally the MVP data being unfiltered will have internal waves (as acknowledged) which may misleadingly increase the lateral buoyancy gradient used for Ri_g. As an aside, might the striped nature of Fig. 5 be due to internal waves?

For the mesoscale, the argument largely rests on advection using the altimetry derived flows and the Lyapunov exponents. Presumably if the sky was clear enough for such Chl images then SST is also available. This should also be shown in Fig. 6 (or in an equivalent new figure) as it gives greater faith in the analysis. SST does not have the complication of being a reactive tracer. i.e. does the SST field show the same/different matches with the FSLEs? Are theye consistent with the hypothesis? I find the particle tracking back to 25 December uncompelling, particularly given the very interesting idea in the Discussion of island iron being a factor. At first glance, the most striking feature of Fig 6 is a high chl patch on the east side of the island which seems about to be drawn away by the mesoscale flow. This is the basis of the authors' iron suggestion and they need to do it more justice. It would be interesting to see the results of seeding particles over the patch of high Chl next to the island on Jan 13 and running this forward to see

how these waters correspond to those found later in the patch hosting LDB. If there are more satellite images available between Jan 13 and 31 in particular these should also be shown. On a more technical note the authors should discuss the consequences of using 30d integrations for FSLEs given the extent to which the 2d circulation is apparently changing in Fig. 6. How does the match up to tracers change if shorter integrations are used?

I additionally have a number of more minor comments/suggestions:

- Given the issue with the salinities from the CTD I think there needs to be a T/S diagram using MVP data in Supplementary material to reassure the reader that density comparisons between Fig. 2 and 3 are reliable

- The location of the CTDs taken at LDB needs to be indicated on Fig. 2 and 6

- Fig. 3 looks like there might be an issue with quenching as the increase and decrease in surface Chl up to Mar 18 seem to have the expected daily cycle.

- This isn't really relevant to the main question behind the paper but why do the NO3 and PO4 profiles have a maximum around 120m?

- If the hypothesis is of P controlling N fixation which drives the bloom it might be worth doing a scatter plot of PO4 versus Chl in surface waters (taking care with quenching) as a negative correlation would support this.

- There is no Section 2.3.1 – first line , p7

- Figs 2 and 3 need the same colour scale

———————————————————

---

## Referee Comment (RC2) · Anonymous Referee #2 · 25 May 2017

In this manuscript, the authors use hydrographic data as well as remotely sensed data to describe the evolution of a phytoplankton bloom which was observed during an oceanographic cruise. In-situ data used in this study comes from 3 stations, S12, LDB and SD13, which were taken from Mar 11 thru Mar 21 2015. The authors conclude that the mesoscale eddy field is responsible for the horizontal advection of the bloom and do not find submesoscale motions to be relevant in the study region during that period, as diagnosed from the gradient and the balanced Richardson number.

The manuscript is well written and describes in detail the analysis and how the authors

base their conclusions. At times, though, it reads much like a cruise report. I believe the authors could be more concise and to the point.

My main concern about the manuscript is what is exactly new in this study. The authors rule out the role of submesoscale motions in the horizontal distribution of the bloom. However, the main role of such motions in oligotrophic regions would be to ignite surface chlorophyll blooms by supplying limiting nutrients to the surface. This would occur, by definition, at the onset of the blooms. The in-situ sampling in the study took place from Mar 11 to Mar 25, when the bloom, as seen from the satellite images (Fig. 6) was relatively mature.

The authors point out that it probably started on the previous December in the vicinity of an island. They are probably correct that some type of island-induced fertilization occurred, thus alleviating nutrient limitation (Dore et al. 2008), with chaotic advection transporting material over long distances, as shown previously (Rypina et al 2010). However, with the evidence shown it is not possible to infer if submesoscale processes were at work at the beginning of the bloom. Also, Law et al. 2011 report high rates of nitrogen fixation in an oligotrophic region after the passage of a tropical cyclone, which supposedly fertilized the ocean prior to a bloom. Strong winds may or may not be important for the ignition of the observed bloom, but the authors do not mention anything about it. The horizontal evolution of the bloom is most likely controlled by mesoscale currents, as shown in previous studies (Calil et al. 2011).

The authors claim to use a formulation from Thomas et al. 2013, based on the balanced Richardson number, to determine "how submesoscale the observed velocity shear is". However, the criteria described in Thomas et al. 2013, as seen by the pie chart in their Fig. 1, characterizes the flow as stable or unstable to a number of instabilities. Moreover, it considers the relative vorticity of the flow field. Therefore, while I don't think submesoscale processes were at play during the survey, this diagnostic by itself is not fully accurate for the purposes intended in this study and may be misleading for readers.

As a general comment, it has long been recognized that subtropical gyres, despite their low biomass, are far from being "oceanic deserts" (Emerson et al. 1997) as they are responsible for approximately half of the export of organic carbon of the oceans.

An additional comment: the authors tend to use sentences such as "investigators often espouse the assumption that" or "which are what most investigators focus on". These sentences, without specific references are vague and unfit for a scientific paper. The authors should explicitly cite the works or assumptions they are supposedly challenging or simply remove these sentences.

References

Calil, P. H. R., S. C. Doney, K. Yumimoto, K. Eguchi, and T. Takemura (2011), Episodic upwelling and dust deposition as bloom triggers in low‐nutrient, low‐chlorophyll regions, J. Geophys. Res., 116, C06030, doi:10.1029/2010JC006704

Dore, J., R. Letelier, M. Church, R. Lukas, and D. Karl (2008), Summer phytoplankton blooms in the oligotrophic North Pacific Subtropical Gyre: Historical perspective and recent observations, Prog. Oceanogr., 76, 2–38.

Emerson, S., P. Quay, D. Karl, C. Winn, L. Tupas, and Mo Landry. "Experimental determination of the organic carbon flux from open-ocean surface waters." Nature 389, no. 6654 (1997): 951-954.

Law C.S. , Woodward E. M. S. , Ellwood M. J. , Marriner A. , Bury S. J. , Safic K. A. , (2011), Response of surface nutrient inventories and nitrogen fixation to a tropical cyclone in the southwest Pacific, Limnology and Oceanography, 56, doi: 10.4319/lo.2011.56.4.1372.

Rypina, Irina I., Lawrence J. Pratt, Julie Pullen, Julia Levin, and Arnold L. Gordon. "Chaotic advection in an archipelago." Journal of Physical Oceanography 40, no. 9 (2010): 1988-2006.

Thomas, L. N., Taylor, J. R., Ferrari, R., and Joyce, T. M.: Symmetric instability in

the Gulf Stream, Deep Sea Research Part II: Topical Studies in Oceanography, 91, 96–110, 2013.

---

## Author Comment (AC1) · 20 Jun 2017

**Response to Reviewer 1**
for "The Fate of a Southwest Pacific Bloom: Gauging the impact of submesoscale vs. mesoscale circulation on biological gradients in the subtropics"

by Alain de Verneil, Louise Rousselet, Andrea M. Doglioli, Anne A. Petrenko, and Thierry Moutin

We thank Anonymous Reviewer 1 for their time and effort in formulating their review of the manuscript. Below we reproduce the reviewer's response and address their concerns along the way.

*The topic of how primary production may be sustained by mesoscale and submesoscale circulations, particularly in the low nutrient subtropical gyres, is one of considerable broad interest. Yet it is an area where observations remain relatively few, particularly compared to modelling studies despite several of the latter indicating a major role for such physical processes. Motivated by encountering an unexpected phytoplankton bloom in such a region, the authors use a combination of in situ data and remote sensing to explore the potential role of the physical circulation. Their conclusion - that vertical nutrient flux from submesoscale processes is an unlikely control and that it is largely a 2D phenomenon with the mesoscale field advecting water against the mean flow - is one that seems best supported by the data. This is a very interesting result, particularly with the corollary that the mesoscale flow may be drawing iron away from the islands to trigger the bloom. However, there are a few aspects of the paper that I think need addressing before the manuscript can be published.*

*The paper essentially considers two possibilities: submesoscale vertical movement and mesoscale horizontal movement. The reader needs to have faith that both options have been thoroughly tested before the conclusion can be reached.*

*For the submesoscale, the largely horizontal structure of density and*

*discontinuities between surface and deep Chl are pretty convincing. I find the Richardson number argument less so given that the majority of submesoscale motions are confined in the surface boundary layer which is poorly sampled judging by Fig. 5. The authors should acknowledge this.*

**Response to Comment**
Yes, indeed, the surface layer where one would expect most submesoscale motions is poorly sampled, since the mixed layer, according to our definition, averaged around 20 dBar for both the MVP and CTD datasets. Additionally, some data were cut off from this surface layer so that a direct comparison could be made to Ri using the ADCP data. The problems with the horizontal resolution of these features was mentioned in Sect. 2.4, Pg. 6, lines 26-29, as well as the Discussion. We have added the following changes (in bold) to the latter, where the possibility that the MVP survey did not fully resolve the features of interest is mentioned:

> Discussion, Sect. 4.2, Page 13, Line 24:
> If most submesoscale structures are expected to be in the mixed layer, which throughout this dataset was near 20 m, then the sub-kilometer $R_D$ (here found to <200m) would not have been resolved by the MVP survey **since MVP horizontal resolution is $\sim$ 2 km. Besides problems of horizontal resolution, the shallow mixed layer also precluded complete vertical resolution by the underway MVP. As a result, though the summertime conditions present during the surveys lend support to reduced submesoscale circulation, the very same conditions make it difficult to state with confidence that the $Ri$ and $Ri_g$ methodology is entirely conclusive.** However, since the bloom of interest in these surveys spanned the top 40 m, and covered hundreds of kilometers in horizontal extent, these small features, should they have existed, would not have impacted the full depth range of the bloom, nor would they have significantly affected the horizontal advection of the entire bloom.

*For clarity, Fig 5 should also have its colour scale changed so that*

*the blue-yellow transition is centred on zero - the value of interest.*

**Response to Comment**

We have changed Fig 5 in accordance with this suggestion.

*Additionally the MVP data being unfiltered will have internal waves (as acknowledged) which may misleadingly increase the lateral buoyancy gradient used for $Ri_g$. As an aside, might the striped nature of Fig 5 be due to internal waves?*

**Response to Comment**

Regarding the aside, yes, the striped nature of Ri in Fig. 5 was most likely due to internal waves. In particular, near-inertial oscillations were observed during OUT-PACE and are also likely a part of the ADCP data set during the MVP transects. Therefore, we looked into how filtering to remove these effects alters our results, which is summarized below.

The reviewer is astute in noting that by not horizontally filtering the density, an aliased internal wave may increase the horizontal buoyancy gradient, thus biasing $Ri_g$ toward smaller values. For comparison, below is what Fig. 5 looks like when density is horizontally filtered, and we have also provided histograms of $Ri_g$'s distribution for the unfiltered and filtered treatments over the top 50 m depth. Due to the homogeneity of density, the median decorrelation length scale (as judged by zero-crossings of the auto-correlation function) for density in the upper 50 m of T4 was approximately 34 km ($\sim$ 18 observations). Since this length scale is a significant fraction of the full-depth Rossby radius alluded to in the text (and in itself somewhat suggests the natural scale being mesoscale), below we present the 34 km filter (right panels) alongside a shorter 10 km filter treatment ($\sim$ 5 observations) for comparison (middle panels).

[Figure]

As one would expect, the filtered $Ri_g$ has larger values (ie weaker vertical shear due to horizontal density structures), and only a few observations are at the value of 1 (0 in log-scale). The differences between the 10 km and 34 km filtering are virtually non-existent. For the purposes of this manuscript, since even with a biased signal the conclusion was that submesoscale shear was inconsequential, this further substantiates our claims. Therefore, we propose to keep the unfiltered Figure 5, but to add the above figure as part of the Supplementary material as Fig. S3 (Figures S1-S2 will concern the T-S figure requested). Additionally, we have added to the Materials and Methods section to refer to this sensitivity to filtering (changes in bold):

Materials and Methods, Sect. 2.2.2, Pg. 4, Line 30:
As discussed below (Sect. 2.4), the stratification was such that, even at 2 km resolution, density structures associated with balanced currents near the surface might be missed, so no additional filtering was applied to density in the MVP dataset despite possible aliasing of internal waves. **The $Ri_g$ number analysis detailed in Sect. 2.4 can be sensitive to these aliased waves, and possibly bias the results. While, in general, this processing step should be considered, sensitivity analysis shown in the Supplementary Material (Fig. S3) revealed that the lack of filtering with the current dataset did not affect the resulting conclusions.**

*For the mesoscale, the argument largely rests on advection using the altimetry derived flows and the Lyapunov exponents. Presumably if the sky was clear enough for such Chl images then SST is also available. This should also be shown in Fig. 6 (or in an equivalent new figure) as it gives greater faith in the analysis. SST does not have the complication of being a reactive tracer. i.e. does the SST field show the same/different matches with the FSLEs? Are they consistent with the hypothesis?*

**Response to Comment**

As the reviewer has surmised, yes, SST data are available, and indeed SST is a more reliable tracer than Chl. Beyond the biological emphasis of this manuscript, Chl was used instead of SST because SST contained fewer visible gradients. We believe this is due to the fact that regional summertime heating at these latitudes is strong. The SST equivalent of Fig. 6 is provided below:

[Figure]

The FSLEs that are the focus for the current manuscript (in areas of high chl-a) overlap regions of homogeneous SST. However, for certain dates FSLEs can be seen to correspond with SST fronts. Jan. 31, for example, indicates a recirculation of water south of the bloom's exit from the island region with cooler values to the South. Additionally, FSLEs and cooler water to the South are well-aligned in the last three panels (Apr 03 to May 10). Though the SST unfortunately does not contribute to the bloom's narrative, for completeness, we will add the above SST figure as supplementary material Fig. S4, and include the following references in both the Results and Discussion (changes in bold):

> Results, Sect. 3.4, Pg. 9, Lines 20-24:
> The remotely sensed distribution of surface chl-a, calculated FSLEs and ARIANE Lagrangian particle positions, over a period spanning 25 December, 2014 to 10 May, 2015 are shown in Fig. 6. FSLEs and particles are shaded gray and red, respectively, with 10% of the particles randomly selected for plotting in all subpanels. 25 December (Dec., all months hereafter shortened) was chosen as the starting point by visually examining the chl-a dataset for a pre-bloom period, with a bloom 'source' region identified on 13 Jan. centered at 186° E, 20° S. **The temporal evolution of FSLEs and the of Lagrangian particles superimposed on SST is reproduced in Fig. S4 of the Supplementary Material.**

> Discussion, Sect. 4.2, Pg. 14, Line 14:
> Evidently, chl-a is a reactive tracer and undergoes its own evolution, as shown by the shaded areas in Fig. 7b. **By contrast, the temporal evolution of SST, which does not suffer from this deficiency, did not display enough variability in the bloom region to confirm the efficiency of FSLEs or ARIANE particles in representing its advection (Fig. S4 in the supplementary material). This is most likely due to the strong, regional summertime heating that occurs at these latitudes.**

*I find the particle tracking back to 25 December uncompelling, particularly given the very interesting idea in the Discussion of island iron being a factor. At first glance, the most striking feature of Fig 6 is a high chl patch on the east side of the island which seems about to be drawn away by the mesoscale flow. This is the basis of the authors' iron suggestion and they need to do it more justice. It would be interesting to see the results of seeding particles over the patch of high Chl next to the island on Jan 13 and running this forward to see how these waters correspond to those found later in the patch hosting LDB. If there are more satellite images available between Jan 13 and 31 in particular these should also be shown.*

**Response to Comment**

We performed the requested Lagrangian analyses and below we provide the corresponding figures. First, Fig. 6 is re-created with a new particle initialization at the bloom's position near the island group on Jan 13 and advected in a forward time integration. Hereafter, we will refer to this as "Fig-6-Forward". Next, we provided the timeseries of Chl/FSLE/particle figures, starting from Jan 10 (a few days earlier than Jan 13) to Jan 31 with the Mar 06 particle seeding from the manuscript to fill in the gaps not presented in the original Fig. 6.

[Figure]

[Figure]

The additional Chl/FSLE/particle images between Jan 10 to Jan 31 document the bloom's exit from the island group in an eastward direction. The importance of the North-South FSLE barrier is evident during this period. The particles seeded in the bloom on Jan 13 (Fig-6-Forward) are largely absent from the bloom on Mar 06 (the seeding date in the original Fig. 6) and on Mar 21, the MVP sampling date. While disappointing for our argument, it is not surprising that chaotic particle trajectories are not strictly accurate two months after their initialization. This can be due to the reactive nature of chl-a, but also to the motions unresolved by our satellite data. The reactive nature of chl-a may be important, in particular for the blooms of this region that are sustained over long time periods by $N_2$ fixation. Symmetric to the lack of particles inside the bloom in Fig-6-Forward after two months, the backward integration shown in the original Fig. 6 produced few particles inside the bloom on Jan 13 near the island source region, also almost two months prior to seeding. If the errors result from unresolved physical motions, these results imply a temporal window beyond which particles have deviated from their "true course", and this is clearly less than two months. The particles in both scenarios, despite their limitations, both show eastward advection over time, which was one of the main results in our analysis. Additionally, while the particle positions are sensitive to the errors in the velocity field, the FSLE structures are relatively robust to these errors (mentioned in Sect. 2.4, Pg. 7, Line 7 with reference), and both particle seeding experiments show their role as flow barriers.

The original Fig. 6 was chosen to show the temporal evolution of the bloom both before its appearance on Jan 13 and after its decline starting in April. While the Dec 25 integration of particles may be uncompelling, the FSLEs are still relevant. Additionally, though there is much interest in following the bloom from its source and identifying its causes, one of the focuses of this work is in comparing which circulation regime creates biological gradients. In order to exemplify the ability of the mesoscale regime to form the gradients observed both by remote sensing and in situ data (e.g. MVP Transect 4), we feel the Mar 06 seeding is more relevant for the exposition of gradient formation (see original Fig. 6g). To prevent figure clutter but to also still show the results shown above, we propose adding both figures to the Supplementary Material as Figures S5 (Fig-6-Forward) and S6, respectively. We have modified the manuscript in the following ways:

Materials and Methods, Sec. 2.4, Pg. 7, Line 18 (additions in bold):

Lagrangian particles were spaced $1/50°$ ($\sim 2$km) apart within the chl-$a$ contour of $0.3$ mg m$^{-3}$. **An additional forward particle experiment initialized on Jan 13, with a localization of the bloom near an island group, was also conducted, and these results are shown in supplementary material Fig. S5.**

Results, Sect. 3.4, Pg. 9, Lines 20-24 (previous changes repeated in italics, additions in bold):

The remotely sensed distribution of surface chl-$a$ with a bloom 'source' region identified on 13 Jan. centered at $186°$ E, $20°$ S. *The temporal evolution of FSLEs and the Lagrangian particles superimposed on SST is reproduced in Fig. S3 of the Supplementary Material.* ***Additional chl-a data, between January 10 and 31, 2015, are also provided in Fig. S6.***

Discussion, Sect. 4.2, Pg. 14, Line 14 (previous changes repeated in italics, additions in bold, deletions with a strikethrough):

Evidently, chl-*a* is a reactive tracer and undergoes its own evolution, as shown by the shaded areas in Fig. 7b. *By contrast, the temporal evolution of SST, which does not suffer from this deficiency, did not display enough variability in the bloom region to confirm the efficiency of FSLEs or ARIANE particles in representing its advection (Fig. S3 in the supplementary material).* **Moreover, the particle positions are not reliable over long timescales. Few particles can be found in the bloom on Jan 13, when it was localized near a group of islands. Conversely, a second initialization experiment on Jan 13 failed to produce many particles in the bloom for Mar 06 and Mar 21 (Supplementary Material, Fig. S5). This limitation may be a result of chl-a being a reactive tracer or of unresolved motions in the altimetry-derived flow field. Therefore, while** the ability of the Lagrangian particles to  remain in the region of interest and to accurately represent elevated chl-a values (mostly above or near the 75th percentile) provides  positive evidence that mesoscale flows were indeed advecting the bloom water **, after around two months the accumulated errors due to unresolved flows make direct inspection of particle position uninformative. The FSLEs, in comparison, do not suffer from this sensitivity.**

Discussion, Sect. 4.2, Pg. 14, Line 18-20 (additions in bold, deletions with a strikethrough):

Firstly, the  *location of the bloom* near an island group *on Jan 13 (Fig. 6b, S5)*  suggests a possible island effect in the ignition of the bloom. **Despite the fact the bloom's beginning was not captured by in situ data, we still suggest a mechanism responsible for causing the bloom.** Considering that $N_2$ fixation drove new production, and nearby stations SD12 and SD13 had detectable phosphate levels, alleviation of another necessary and limiting nutrient, iron, was possibly at work.

Discussion, Sect. 4.2, Pg. 14, Line 24-25 (additions in bold, deletions in strikethrough):

Secondly, the shifting FSLEs and **both** Lagrangian particle **experiments (Figs. 6, S5, and S6)**  demonstrate the general eastward advection of the bloom from its localized island source in Fig. 6b until its easternmost position in Fig. 6g.

*On a more technical note the authors should discuss the consequences of using 30d integrations for FSLEs given the extent to which the 2d circulation is apparently changing in Fig. 6. How does the match up to tracers change if shorter integrations are used?*

**Response to Comment**

The position of FSLEs, especially the strongest features, are fairly robust to the integration timescales. Long integration periods, in general, resolve smaller-scale features. Below we provide the FSLEs using 10, 15, 20, 25, and 30d integrations for Jan 31, when the bloom advected away from the island group and experienced strong North-South shear.

[Figure]

The largest change is between 20 and 25 days. Past this point, at 30 days, weaker structures appear, but are removed by the 0.15 d$^{-1}$ threshold. Hence, in our case, there is no point in integrating with longer periods than 30 days, and the sensitivity of the FSLEs in our dataset should be minimal. We have added mentioned these sensitivity tests in the materials and methods:

Materials and Methods, Sec. 2.4, Pg. 7, Lines 6-7 (additions in bold):

The robustness of FSLE calculations to small-scale errors in velocity fields has been previously studied (Cotte et al., 2011). **The 30 day integration timescale we have chosen is likewise robust. Sensitivity analyses (not shown) indicate the strongest features are resolved with 10-15 day integrations, with finer detail emerging over 25-30 day integrations. The smallest structures are removed by a 0.15 day$^{-1}$ threshold for the analysis in this study.**

*I additionally have a number of more minor comments/suggestions:*

*- Given the issue with the salinities from the CTD I think there needs to be a T/S diagram using MVP data in Supplementary material to reassure the reader that density comparisons between Fig. 2 and 3 are reliable.*

**Response to Comment**

We have plotted a T/S diagram of the MVP data for the calibration cast with SD13, shown below. Another CTD cast taken mid-way during other MVP sampling not presented in this manuscript is also shown. The first calibration casts were 3 km and 41 min apart, and the second set was 0.5 km and 53 min. apart.

[Figure]

The variability in salinity is obvious, especially near the surface. This appears as a consequence of the fact that the sound speed is largely sensitive to temperature, and not salinity. As a result, the sound speed inversion to get salinity is relatively indeterminate. One would also expect variability to be greater at the surface, so this is not entirely surprising, but highlights that while the MVP allows for rapid surveying, this comes at the cost of increased signal variability vis a vis CTD rosette sampling. However, since the greater part of stratification was due to temperature, the variability due to salinity presents itself as small-scale salinity spikes in density that, while minimized in our post-processing, are still present (see density profiles below). The three profiles are off-set by 0.25 kg m$^3$ each.

[Figure]

Once the density is re-ordered, the mis-fit between the calculated density of the MVP and the CTD is small enough ($r^2$ of 0.99) that the reader should be reassured of its utility for the plots and calculations. We propose to add the two above figures in the Supplementary material, and have made reference to them in the manuscript:

Materials and Methods, Sect. 2.2.2, Pg. 4, Lines 22-24 (additions in bold):

Due to technical difficulties onboard the ship, the conductivity sensor was swapped for a sound velocity sensor. In order to calculate salinity, the roots of the sound speed equation from Chen and Millero (1977) were matched with Mackenzie's linear approximation (1981). Sound speed and temperature data were lag-corrected to reduce salinity spiking. **The variability in calculated salinity from the soundspeed is larger than that calculated with conductivity (Fig. S1), but due to the greater contribution of temperature to stratification the resulting density profiles compare well with the CTD (Fig. S2, $\rho$ =0.998, r$^2$ =0.996).**

*- The location of the CTDs taken at LDB needs to be indicated on Fig. 2 and 6*

**Response to Comment**
We have added the LDB CTD locations to both Fig. 2 and 6. However, due to the small spatial range of the CTD locations in relation to the area mapped, these locations all overlap and appear as one point.

*- Fig. 3 looks like there might be an issue with quenching as the increase and decrease in surface Chl up to Mar 18 seem to have the expected daily cycle.*

**Response to Comment**
The reviewer's observation of a daily cycle, most likely involving non-photochemical quenching, was noted by the authors as well. This effect would be problematic if the conclusions relied more sensitively upon the quantitative values of Chl observed. However, due to the large-scale change in surface Chl after Mar 18, the authors feel that the uncorrected contribution due to quenching does not impact the results. Nevertheless, to acknowledge the Reviewer's point, the quenching effect has been noted in the Materials and Methods, Results, and Discussion, as follows:

Materials and Methods, Sect. 2.2.1, Pg.3, Lines 26-27 (additions in bold):
Chl-$a$ fluorescence was calibrated to chl-$a$ extractions taken from the bottle samples throughout the cruise. **No corrections were made for the daily oscillations due to non-photochemical quenching.**

Results, Sect. 3.1.2, Pg. 8, Lines 12-13 (additions in bold):
From 18 March onward, the surface concentration of chl-$a$ also decreased, whereas the chl-$a$ max concentration increased and began to resemble a typical DCM distribution. ***Oscillations in the surface chl-a during the first half of the timeseries appear, likely due to non- photochemical quenching.***

Discussion, Sect. 4.1, Pg. 12, Line 17 (additions in bold):
Station LDB's CTD timeseries also showed the decrease of surface chl-$a$ and the new formation of a DCM near 80 m. ***Fluctuations in the surface chl-a, a possible artifact of non-photochemical quenching, were small in relation to the large change in chl-a that occurs between the first and second halves of the timeseries.***

*- This isn't really relevant to the main question behind the paper but why do the NO3 and PO4 profiles have a maximum around 120m?*

**Response to Comment**
The maximum in nutrients in SD12 was noticed by the authors, as well. This local maximum in nitrate and phosphate is also reproduced in silicate. Additionally, a local oxygen minimum is also found at this depth. As a result, it is not likely a measurement error. Conversely, the temperature and salinity for SD12 don't show any discontinuities or anomalous trends. At depths greater than the 200m limit shown in Fig. 4, the usual trends of increasing nutrients are re-established for SD12. We interpret this local departure from a "classic" nutrient profile to be an intrusion at this depth of an another water mass, and without further data it is difficult to interpret its source. The water mass analysis of the entire OUTPACE

transect will be the scope of another paper from the special issue (Fumenia et al., this issue). As the reviewer notes, this does not directly impact the conclusions of the paper concerning the upper layer, but indeed it is not a mistake.

*- If the hypothesis is of P controlling N fixation which drives the bloom it might be worth doing a scatter plot of PO4 versus Chl in surface waters (taking care with quenching) as a negative correlation would support this.*

**Response to Comment**
The author's suggestion to test P control on N fixation, and thus growth, by plotting PO4 and Chl for a negative correlation is a good one. Unfortunately, the extremely low values of PO4 in the surface waters of LDB were below the measurement threshold, and so no reliable scatter plot can be made.

*- There is no Section 2.3.1  first line, p.7*

**Response to Comment**
Thank you for noting this, it has been corrected to just say Sect. 2.3 in the text.

*- Figs 2 and 3 need the same colour scale*

**Response to Comment**
Thank you again for seeing this, the change has been implemented.

---

## Author Comment (AC2) · 20 Jun 2017

**Response to Reviewer 2**

for "The Fate of a Southwest Pacific Bloom: Gauging the impact of submesoscale vs. mesoscale circulation on biological gradients in the subtropics"

by Alain de Verneil, Louise Rousselet, Andrea M. Doglioli, Anne A. Petrenko, and Thierry Moutin

We thank Anonymous Reviewer 2 for their time and effort in both reading the manuscript and writing their review. Below the review is reproduced with our responses to the concerns raised.

*In this manuscript, the authors use hydrographic data as well as remotely sensed data to describe the evolution of a phytoplankton bloom which was observed during an oceanographic cruise. In-situ data used in this study comes from 3 stations, SD12, LDB and SD13, which were taken from Mar 11 thru Mar 21 2015. The authors conclude that the mesoscale eddy field is responsible for the horizontal advection of the bloom and do not find submesoscale motions to be relevant in the study region during that period, as diagnosed from the gradient and the balanced Richardson number.*

*The manuscript is well written and describes in detail the analysis and how the authors base their conclusions. At times, though, it reads much like a cruise report. I believe the authors could be more concise and to the point.*

*My main concern about the manuscript is what is exactly new in this study. The authors rule out the role of submesoscale motions in the horizontal distribution of the bloom. However, the main role of such motions in oligotrophic regions would be to ignite surface chlorophyll blooms by supplying limiting nutrients to the surface. This would occur, by definition, at the onset of the blooms. The in-situ sampling in the study took place from Mar 11 to Mar 25, when the bloom, as seen from the satellite images (Fig. 6) was relatively mature.*

*The authors point out that it probably started on the previous December in the vicinity of an island. They are probably correct that some*

*type of island-induced fertilization occurred, thus alleviating nutrient limitation (Dore et al. 2008), with chaotic advection transporting material over long distances, as shown previously (Rypina et al 2010). However, with the evidence shown it is not possible to infer if submesoscale processes were at work at the beginning of the bloom. Also, Law et al. 2011 report high rates of nitrogen fixation in an oligotrophic region after the passage of a tropical cyclone, which supposedly fertilized the ocean prior to a bloom. Strong winds may or may not be important for the ignition of the observed bloom, but the authors do not mention anything about it. The horizontal evolution of the bloom is most likely controlled by mesoscale currents, as shown in previous studies (Calil et al. 2011).*

**Response to Comment**

In this section the reviewer first details the expected role of submesoscale vertical motion in starting a bloom, and that no in situ data in this study can corroborate submesoscale motion in the bloom's ignition. In the subsequent paragraph, the relative roles of island-induced fertilization, chaotic mesoscale advection, and strong wind forcing are mentioned with references to highlight the purported lack of novelty. Framed in this manner, we understand the reviewer's opinion and here we will better communicate the novelty in this work, and what hypotheses are being tested that contribute to the scientific literature.

Firstly, the reviewer is right in that the vertical motions due to submesoscale dynamics are probably of most interest to biologists, given their enhanced magnitudes relative to mesoscale motion (Mahadevan and Tandon, 2006). However, horizontal motions are also important for biological applications by influencing patch dynamics. Biological gradients are important because they can be hotspots of predation and other trophic interactions, where in terrestrial environments these are called 'edge effects' (Harris, 1988). Therefore, it matters which circulation regime is advecting a patch of bloom water. Both mesoscale and submesoscale regimes are expected to stir and strengthen gradients in a forward tracer variance cascade (Klein et al., 1998), but submesoscale motions would concentrate more of that variance at even smaller scales. Having more patches (and by definition more biological gradients at their boundaries) at small scales means greater opportunity

for these biological dynamics. Therefore, we have added the following to our introduction to better highlight this focus on horizontal motions and why it is important to diagnose submesoscale vs mesoscale regimes (and also remove unnecessary sentences mentioned later in the review):

> Introduction, Sect. 1, Page 2, Line 4 (additions in bold italics, deletions shown in strikethrough):
>
> Marine biological communities at any moment reflect a time-integration of the many complex interactions that occur both within the community and with the physical environment (Longhurst, 2010).  ***An important structuring mechanism of biological communities is the presence of gradients. In the terrestrial and conservation biology literature, these impacts are dubbed 'edge effects' (Harris, 1988), and have important implications for predation processes and species survival. Horizontal patch edges and biological transitions in the Ocean are liable to being advected by the surface circulation. Thus, it is necessary to identify the character of the flows that shape these horizontal gradients.***

We agree with the reviewer that it would be very interesting to know what happened at the beginning of the bloom. Nevertheless, as the reviewer noted, the proceeding of cruise events meant that we have no in situ data for the bloom at this period. Consequently, we focused on the "Fate" and not "Birth" of a Southwest Pacific bloom, as granted in the title of the paper. However, we do go into some detail regarding what can be inferred about the bloom's biogeochemical starting point in the Discussion. Therefore, as a means of better exploring the possible explanations for the bloom's origins, we have taken into account the mechanisms raised by the reviewer and tested whether these can be attributable to the bloom.

Since *Trichodesmium* blooms are rare (Westberry and Siegel 2006), they need to be treated on a phenomenological basis, as in the literature cited by the reviewer (Calil et al. 2011, Law et al. 2011). We thank the reviewer for pointing out these previous studies. Investigating these alternative mechanisms for the bloom in this manuscript can provide a further means of hypothesis testing on a rare event, which is a useful contribution to the scientific community.

Calil et al. (2011), cited by the reviewer, diagnosed upwelling motions due to mesoscale frontogenesis/frontolysis as estimated by the Omega equation (Hoskins et al., 1978), albeit with an alternate formulation and simplifying assumptions so that satellite altimetry data can be used. We have taken their approach, and used their Eqn. 2, ignoring the second deformation term, and calculated the right hand side of the equation, eg:

$$2f_0 \frac{\partial}{\partial z}(\mathbf{v}_g \cdot \nabla_h \zeta_g)$$

simplified further to:

$$2f_0 \mathbf{v}_g \cdot \nabla_h \zeta_g$$

Whether this quantity is negative or positive will imply a upwelling or downwelling velocity, respectively. Below is a figure of the currents, vorticity, and this Omega equation term for Jan 13, 2015, as the bloom is about the leave the island group. Each quantity is shown with its histogram directly below it ($log_{10}$ of absolute magnitude for vorticity and Omega term).

[Figure]

These distributions are similar for other days (figures and data can be provided, if requested). Of note, the omega equation upwelling/downwelling term has minima/maxima with $O(10^{-14})\text{s}^{-3}$, whereas in Calil et al. (2011), the values were three orders of magnitude higher. Additionally, the frontogenesis/frontolysis regions in Calil et al. (2011) coincided with low SST anomalies, which are not seen in our dataset (a SST version of Fig. 6 is to be added to the Supplementary Material in response to Reviewer 1).

Moreover, the upwelling cited in the 2008 bloom of Calil et al. (2011) advected nutrients from a 40 m deep mixed layer, where climatological data from station ALOHA indicate nutrient reservoirs exist. The in situ data from OUTPACE, with a shallower mixed layer near 20 m, show that phosphate, one of the limiting nutrients for nitrogen fixation, was not present immediately below this layer. Instead, the phosphocline was observed near 80 m depth for both LDB and SD12, the non-bloom station in the same region. High phosphate near the surface was instead only observed to the East, in SD13 associated with the subtropical gyre. The weak forcing, lack of SST gradients, and low phosphate in the upper 80 m help rule out this mechanism as judged by remote sensing data.

Calil et al. (2011) also highlight the requisite condition of 25°C for bloom ignition in the 2010 North Pacific bloom. The advancement of the 25° isotherm is invoked to explain the apparent eastward propagation of the large, contiguous bloom. Eastward mesoscale advection is discounted due to lack of evidence from

satellite altimetry data. For the OUTPACE bloom, the SST near the island group immediately before the bloom was mostly above 25°C, though small regions of water below 25°C can be found (see Jan 2015 subpanels of Chl below, with black dots for values $\leq$ 25°C). After Jan 10, however, only a very limited region is below this threshold, and yet only a subset of all the remaining warm water experienced a bloom.

[Figure]

Therefore, while sufficiently warm waters may be ultimately better for diazotroph physiology (in particular *Trichodesmium*), our data support the notion that it is necessary but not sufficient to invoke a bloom. Calil et al. 2011 also noted this, with nutrients such as phosphate and iron being needed.

The reviewer points out the role of tropical cyclones in providing the necessary nutrients for blooms with reference to Law et al. (2011). Yes, wind forcing from strong events such as tropical cyclones can fertilize the ocean and bring about $N_2$ fixation blooms. The value-added altimetry products from CLS/CNES included an Ekman component, derived from the ECMWF ERA INTERIM windstress model. Below, moving left to right we plot the velocities without wind, with wind, and

the differences, respectively, for Jan 13. Immediately below these three panels are histograms of the u, v component and % magnitude ratio of the wind-nowind difference to the magnitude of the no-wind product.

[Figure]

[Figure]

[Figure]

[Figure]

[Figure]

[Figure]

The differences between velocities with wind effects and without were small, and the magnitude differences in velocities averaged around 30% of the total. These results suggest little changes due to wind in the circulation. However, there was a strong wind forcing event within the OUTPACE region, just not near LDB. Cyclone Pam entered the Southwest Pacific in early March, and a drop in SST and increase in Chl followed in its wake. The relevant figures are shown below (the islands are the Vanuatu archipelago, top row is SST, bottom is Chl-a. Color scale is 25 to 31°C and -1.5 to 0 $log_{10}$ Chl-a, respectively):

[Figure]

As can be clearly seen, the storm did indeed have a fertilizing effect. Unlike the LDB bloom, however, the elevated Chl signal did not last for an entire month. Therefore, in the face of an extreme forcing event, the biological response was transient in opposition to the LDB bloom, where there was no large-scale forcing and the bloom was persistent for over two months. While we cannot be certain that nitrogen fixation was a major factor in the storm-induced bloom, the longer timescale for the LDB bloom suggests that ongoing new production (we remind the reader that nitrogen fixation was directly observed in situ at LDB) was an important factor. This further validates our examination of the in situ data which, although not present for the bloom's ignition, can still be used to examine the circulation impacting the continuing new production being created.

In light of the previous examination of mechanisms elicited by our response to the reviewer's comments, we propose to amend our manuscript as follows:

The previous figures, which look at, in turn, frontogenetic/frontolytic forcing, SST thresholds, wind forcing, and a contemporaneous short-lived bloom due to wind forcing, will be added as Supplementary material. In particular, since in response to Reviewer 1 there are currently Figs. S1-6, these will constitute Figs. S7-10. Fig. S7 displays the frontogenetic forcing. Fig. S8 shows the wind-nowind differences. Fig. S9 has the timeseries of SST and Chl following Cyclone Pam. Finally, Fig. S10 shows the position of 25°C water.

The above discussion weighing alternative mechanisms for the bloom's ignition has been added in condensed form in the Discussion, starting where advective fluxes and forcing is considered:

Discussion, Sect. 4.1, Pg. 12, Line 1 (additions in bold):
These data therefore also remove the possibility of a massive
diapycnal mixing event.

***Similar surface blooms in oligotrophic regions have been investigated before, with varying mechanisms to explain their initiation. In particular, upwelling due to mesoscale frontogenesis and wind forcing, are possible causes for surface blooms (Calil et al., 2011, Law et al., 2011). While there are no in situ data during the bloom's appearance in mid-January 2015, sufficient data exist to judge these mechanisms, which would provide advective flux and diapycnal mixing, respectively. Upwelling due to mesoscale frontogenesis can be diagnosed using the Omega equation (Hoskins et al., 1978) with the assumptions employed by Calil et al. (2011) for its use with altimetry data. Calculating this forcing for the OUTPACE bloom resulted in values three orders of magnitude smaller than those for the 2008 bloom of Calil et al. (2011) (Fig. S7). As further comparison, climatological data from station ALOHA in that study place phosphate reservoirs for $N_2$ fixation at 40m depth, shallower than the depths observed during OUTPACE. These results, in addition with the lack of SST gradients one would expect (Fig. S4), make this mechanism unlikely.***

*Another mechanism is strong wind forcing, such as that provided by tropical cyclones. These storms have been shown to fertilize blooms in oligotrophic waters (Law et al., 2011). Using the value-added altimetry dataset with wind component, the impact of wind was evaluated and found to be relatively small (Fig. S8) and could not create deep mixing. By contrast, another region in the OUTPACE domain witnessed the passage of Cyclone Pam in early March, 2015. The satellite imagery before and after its passage corroborate the fertilizing effect of storms in this region (Fig. S9). Whereas the LDB bloom lasted for over two months, this increase in chl-a lasted approximately a month. Therefore, given the lack of strong forcing, a mechanism must be invoked that can produce blooms of greater magnitude and duration than those produced by passing storms.*

Passing reference to the temperature criterion will be added in the subsequent paragraph:

Discussion, Sect. 4.1, Pg. 12, Line 4 (additions in bold italics): Diazotrophs, the organisms responsible for $N_2$ fixation, are normally concentrated in the surface layer *in sufficiently warm water ($\geq$ 25°C in Calil et al., 2011).*  *The LDB bloom was found in the upper surface layer, satellite SST was warmer than the 25°C threshold for its entirety (Fig. S10), and finally* this process was observed directly (Caffin et al., this issue).

*The authors claim to use a formulation from Thomas et al. 2013, based on the balanced Richardson number, to determine "how submesoscale the observed velocity shear is." However, the criteria described in Thomas et al. 2013, as seen by the pie chart in their Fig. 1, characterizes the flow as stable or unstable to a number of instabilities.*

*Moreover, it considers the relative vorticity of the flow field. There-fore, while I don't think submesoscale processes were at play during the survey, this diagnostic by itself is not fully accurate for the purposes intended in this study and may be misleading for readers.*

**Response to Comment**

The reviewer is correct in noting that relative vorticity is included in the conditions described by Thomas et al., 2013 for different instabilities. To clarify, the function of $Ri_g$ in our manuscript is not to primarily search for instabilities. Instability criteria were not the reason that the value of $Ri_g \leq 1$ was chosen to indicate the submesoscale regime. This choice instead comes from other studies (Mahadevan 2016, McWilliams 2016). $Ri_g$ served as a convenient formulation isolating the balanced flow component contributed by submesoscale circulation.

True, relative vorticity is included in the treatment of Thomas et al. (2013), but this is mainly to give mention that inertial instability occurs when anticyclonic relative vorticity becomes stronger than planetary vorticity. Relative vorticity of one sign or another would not affect the value of $Ri_g$. Therefore, if the intention is to diagnose instability conditions, as the Reviewer suggests the relative vorticity needs to be taken into account. Additionally, the $Ri_g$ diagnostic needs to be transformed into the $\phi_{Ri_b}$ variable presented in the pie charts of Fig. 1 in Thomas et al. (2013).

To make these distinctions clear, we have added the following to the Materials and Methods section:

> Materials and Methods, Sect. 2.4, Pg. 6, Line 7 (additions in bold, deletions in strikethrough):
>
> In submesoscale flows, instabilities such as Symmetric Instability (SI) appear when $Ri \leq 1$ (Stone, 1970).  **The submesoscale regime is commonly accepted to begin near $Ri \sim 1$, $Ro \sim 1$ (Mahadevan 2016, McWilliams 2016). In order to diagnose dynamical regimes from in situ data,** we used a formulation from Thomas et al. (2013) to find the geostrophic component of shear, expressed as:

Materials and Methods, Sect. 2.4, Pg. 6, Line 12 (changes in bold italics):

In this paper, we characterized the flow as submesoscale when $Ri_g$ reached this value of 1.

***The $Ri_g$ diagnostic was originally designed for instability criteria. Here, we are not searching for instabilities. The fact that $Ri_g$ solely looks at shear due to buoyancy gradients is useful for considering submesoscale features. In order to more fully investigate the instabilities that are possible in a given dataset, the relative vorticity is required in addition to $Ri_g$ (see Fig. 1 in Thomas et al., 2013; for example, sufficient cyclonic vorticity can make a water column stable to SI below $Ri = 1$), which is out of the scope of this paper.***

*As a general comment, it has long been recognized that subtropical gyres, despite their low biomass, are far from being "oceanic deserts" (Emerson et al. 1997) as they are responsible for approximately half of the export of organic carbon of the oceans.*

**Response to Comment**

We agree with the reviewer, that though subtropical gyres have low biomass, biological rates in the region are high and contribute to an appreciable fraction of organic carbon export. We have removed this characterization from the Introduction.

*An additional comment: the authors tend to use sentences such as "investigators often espouse the assumption that" or "which are what most investigators focus on." These sentences, without specific references are vague and unfit for a scientific paper. The authors should explicitly cite the works or assumptions they are supposedly challenging or simply remove these sentences.*

**Response to Comment**

The following sentences have been removed (in strikethrough):

Introduction, Sect. 1, Pg. 2, Lines 5-8 (Same deletion as shown in Pg. 3 of this document):

Discussion, Sect. 4.2, Pg. 14, Lines 29-32:

This was possible due to the bloom occurring in water no associated with the coherent, elliptic structures that move west. Instead, the bloom occurred in water outside these structures, with tortuous trajectories hyperbolic in nature (Kirwan et al., 2003 **, Rypina et al., 2010**)

**Additional References (not cited in original manuscript or Reviewer's submission)**

Harris, L. D. (1988). Edge effects and conservation of biotic diversity. Conservation Biology, 2(4), 330-332.

Hoskins, B. J., Draghici, I., & Davies, H. C. (1978). A new look at the $\omega$equation. Quarterly Journal of the Royal Meteorological Society, 104(439), 31-38.

Klein, P., Treguier, A. M., & Hua, B. L. (1998). Three-dimensional stirring of thermohaline fronts. Journal of marine research, 56(3), 589-612.

McWilliams, J. C. (2016, May). Submesoscale currents in the ocean. In Proc. R. Soc. A (Vol. 472, No. 2189, p. 20160117). The Royal Society.

Westberry, T. K., & Siegel, D. A. (2006). Spatial and temporal distribution of Trichodesmium blooms in the world's oceans. Global Biogeochemical Cycles, 20(4).